# TABULARBERT: Binning-Based Self-Supervised Learning for Tabular Representation

**Beomjin Park** [* 1]   **Seunghwan An** [* 2]   **Sungchul Hong** [3]   **Hosik Choi** [4]

## Abstract

Tabular data is one of the most fundamental and widely used formats for representing structured information. Classical machine learning algorithms continue to achieve substantial success in extracting predictive patterns and constructing accurate models from structured data; however, representation learning approaches that extend language-model-based methods to the tabular setting have opened new opportunities. Nevertheless, conventional tokenization procedures and token embedding mechanisms are not well-suited to numerical variables, as they fail to preserve key numerical properties, including proximity structure and ordinal relationships. To address this limitation, we propose TABULARBERT, a Transformer-based model that discretizes numerical variables via binning-based tokenization and learns representations that account for numerical proximity and ordinal information while capturing conditional dependencies among variables through masked self-supervised pretraining. We empirically demonstrate the effectiveness and interpretability of the proposed approach, highlighting the benefits of language-model-based representation learning in the tabular domain.

## 1. Introduction

Tabular data remains one of the most widely used formats for structured information, continuing to play a central role across domains such as medicine, bioinformatics, finance, and business despite the growing prominence of unstructured and semi-structured data.

Traditional machine learning methods have been highly effective on tabular prediction tasks. In particular, tree-based gradient boosting models—XGBoost (Chen & Guestrin, 2016), LightGBM (Ke et al., 2017), and CatBoost (Prokhorenkova et al., 2018)—consistently perform strongly on tabular benchmarks and often outperform deep learning in both accuracy and efficiency (Grinsztajn et al., 2022). Their strong performance is commonly attributed to implicit feature selection and the ability to capture irregular, localized patterns through recursive partitioning, which discretizes the input space into a collection of regions and can improve robustness to outliers and distribution shifts (Grinsztajn et al., 2022).

Motivated by the success of large language models, adapting language-model-based representation learning to tabular data has recently become an active area of research. Transformer architectures that treat features as tokens can learn contextualized embeddings and capture feature interactions via self-attention over feature-wise token embeddings (Huang et al., 2020; Somepalli et al., 2021). A key challenge, however, is to represent numerical variables under tokenization while preserving their proximity and ordinal structure, which standard text tokenization schemes do not naturally retain.

To leverage the advantages of discretizing the input space and address the challenges of tokenizing numerical variables, we propose TABULARBERT, which tokenizes numerical variables via binning and learns representations that account for numerical proximity and ordinal relationships while capturing feature interactions. TABULARBERT regularizes token embeddings with a fused-type penalty to encourage smoothness across adjacent bins, and introduces Wasserstein-based and reconstruction objectives to promote distance-aware learning over ordered tokens and mitigate the loss of within-bin information due to discretization. The model is pretrained with masked language modeling (MLM) to learn conditional dependencies among features, then fine-tuned for downstream predictive tasks.

The main contributions of our work are summarized as

---

[*]Equal contribution [1]Department of Information and Statistics, Gyeongsang National University, Jinju, South Korea [2]Department of Information and Telecommunication Engineering, Incheon National University, Incheon, South Korea [3]Department of Information Statistics, Chungbuk National University, Cheongju, South Korea [4]Graduate School, Department of Urban Big Data Convergence, University of Seoul, Seoul, South Korea. Correspondence to: Hosik Choi <choi.hosik@gmail.com>.

*Proceedings of the 43$^{rd}$ International Conference on Machine Learning*, Seoul, South Korea. PMLR 306, 2026. Copyright 2026 by the author(s).

follows:

- We present TABULARBERT, a Transformer-based model tailored to tabular data that leverages the BERT encoder architecture and MLM to learn conditional dependencies among variables while accounting for numerical structure in the learned representations for downstream tasks.

- We propose a learnable token embedding mechanism with fused-type regularization to encourage smoothness across adjacent bins, and formulate Wasserstein-based and reconstruction objectives that account for ordinal structure in contextualized embeddings while mitigating the loss of within-bin variation from discretization.

- We conduct extensive empirical evaluations on multiple real-world datasets and show that TABULARBERT achieves strong performance compared with both traditional tree-based methods and tabular deep learning approaches.

## 2. Related Work

### 2.1. Binning-Based Representations

Several studies have investigated discretization and tokenization schemes for numerical variables. Gorishniy et al. (2022) introduced piecewise linear encoding (PLE), which partitions numerical values into predefined intervals (bins) and encodes each value within an interval using a linear function. PLE is designed to preserve the ordinal structure of the original values by inducing embeddings whose geometry can reflect relative positions across bins, and it can be integrated effectively with Transformer-based architectures for tabular learning. However, PLE preserves ordinal information through a structured deterministic encoding, rather than by learning embeddings whose geometry is refined during training.

Other notable directions include TabMT (Gulati & Roysdon, 2023) and MaCoDE (An et al., 2025). TabMT discretizes continuous variables via clustering and adapts BERT-style masked language modeling to tabular data, demonstrating that discretization combined with MLM can effectively capture cross-field dependencies. Building on this perspective, MaCoDE redefined MLM as histogram-based conditional density estimation. In this framework, normalized numerical variables are discretized into bins and modeled through classification losses. This formulation has been shown to work well for tabular data synthesis and imputation, suggesting that combining discretization with MLM can capture conditional structure in tabular distributions.

Instead of directly encoding input data into embedding vectors, Lee et al. (2024) proposed an autoencoder-based self-supervised framework in which binned numerical values

are used for MLM pretraining, followed by encoder fine-tuning with a task-specific prediction head. To further guide the learning process, they introduced the BinRecon and BinXent losses, which supervise the decoder to reconstruct ordinal or nominal bin indices, thereby encouraging the latent representations to retain bin-level information.

TP-BERTa (Yan et al., 2024) also tokenizes numerical variables through target-aware discretization and adapts pretrained language models to tabular prediction. It learns magnitude-aware representations via supervised pretraining, highlighting the role of discretized numerical tokens in language-model-based tabular learning. Nevertheless, when numerical variables are mapped to bin-level representations, preserving ordinal structure among discretized values remains an important issue.

### 2.2. Transformer Architecture for Tabular Predictive Tasks

Leveraging Transformer architectures for tabular predictive tasks has been an active area of research. SAINT (Somepalli et al., 2021) introduces a hybrid attention mechanism that combines inter-sample attention with conventional self-attention over embedded tabular features. The model is pretrained in a self-supervised manner using an objective that promotes discriminative latent representations across instances, together with a denoising loss that reconstructs the original inputs. After pretraining, SAINT is fine-tuned for downstream prediction using the representation of the `[CLS]` token through a multi-layer perceptron (MLP) head.

Other Transformer-based models have also been proposed for tabular data. FT-Transformer (Gorishniy et al., 2021) proposes a simplified Transformer architecture for tabular learning, where each variable—numerical or categorical—is mapped to a learnable embedding, and a Transformer encoder models inter-feature dependencies. More recently, T2G-Former (Yan et al., 2023) introduces an attention-based architecture that integrates multi-head self-attention with a learnable graph estimator. The graph estimator dynamically constructs relationships between features, which modulate information flow through graph-aware attention, enabling the model to capture both global and local dependencies. These studies demonstrate the effectiveness of Transformer-based architectures for modeling interactions among tabular variables.

## 3. Methodology

In this section, we present our approach to tabular representation learning, which learns numerically structured token embeddings and combines binning-based tokenization with MLM pretraining and downstream fine-tuning.

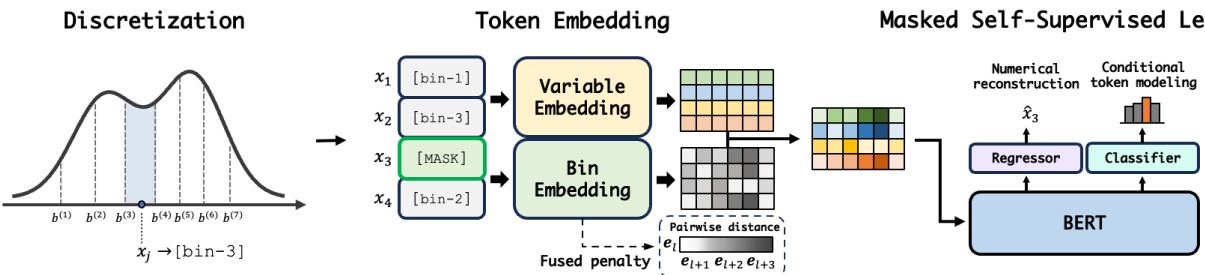

*Figure 1.* Pretraining architecture of TABULARBERT. Numerical variables are discretized into bin tokens, combined with variable and bin embeddings, and trained with masked self-supervised objectives for conditional token modeling and numerical reconstruction.

### 3.1. Binning-Based Tokenization

Let $x_j \in \mathbb{R}$ $(j = 1, \ldots, p)$ denote a real-valued scalar corresponding to the $j$-th tabular variable $X_j$. In contrast to categorical variables, numerical variables are not inherently discretized and thus cannot be directly represented as tokens. To ensure compatibility with language-model architectures, each numerical variable is tokenized into a bin index via a binning procedure.

Formally, for a variable $X_j$ defined over a measurable support interval $[x_j^{\min}, x_j^{\max}]$, we partition the interval into $L_j$ non-overlapping bins:

$$[x_j^{\min}, x_j^{\max}] = \cup_{l=1}^{L_j} B_j^{(l)},$$

where

$$B_j^{(l)} = \begin{cases} [b_j^{(l-1)}, \ b_j^{(l)}), & l = 1, \ldots, L_j - 1 \\ [b_j^{(l-1)}, \ b_j^{(l)}], & l = L_j, \end{cases}$$

with cut-points $x_j^{\min} = b_j^{(0)} < b_j^{(1)} < \cdots < b_j^{(L_j)} = x_j^{\max}$. Each value $x_j$ is then mapped to the bin index in which it falls:

$$x_j \mapsto t_j \in \{1, \ldots, L_j\}.$$

The bin index $t_j$ is used as the token corresponding to the numerical value $x_j$. For a categorical variable with $L_j$ levels, we directly map each category to a unique token in $\{1, \ldots, L_j\}$.

**Cut-point selection**   In binning-based tokenization, cut-points should reflect the underlying data distribution. For highly skewed or heavy-tailed variables, equal-width binning can concentrate most samples into a small number of bins, leaving other bins sparsely populated or even empty. Such imbalanced tokenization reduces resolution in dense regions, since many distinct values are mapped to the same token.

To mitigate this issue, we apply a quantile transformation that maps the marginal distribution of each variable to an approximately uniform distribution. Equal-width binning in this transformed space yields approximately equally populated bins, providing a balanced allocation of tokens across the variable range. In the original feature space, the $l$-th cut-point is given by

$$b_j^{(l)} = F_{X_j}^{-1}(l/L_j),$$

where $F_{X_j}^{-1}$ denotes the inverse marginal cumulative distribution function of $X_j$.

### 3.2. Token Embeddings

The discrete tokens produced by binning-based tokenization are embedded through a learnable embedding layer. However, for numerical variables, the resulting *bin embeddings* do not explicitly preserve the inherent proximity structure of the underlying values. Consequently, embedding vectors associated with adjacent bins may differ substantially, yielding abrupt variations between neighboring token representations. Such discontinuities make the representation sensitive to bin boundaries.

**Fused-type regularization**   To enforce the local smoothness among neighboring bin embeddings, we introduce a fused-type regularization. Let $e_t \in \mathbb{R}^d$ denote the embedding vector corresponding to token $t$ for numerical variables, where $d$ is the embedding dimension. The fused-type regularization is defined as

$$\mathcal{R}_{\text{fused}}(E_{\text{bin}}) = \frac{1}{d^{q/2}} \sum_{l=1}^{L^*-1} \|e_l - e_{l+1}\|_2^q,$$

where $E_{\text{bin}} = (e_{\texttt{[CLS]}}, e_1, \ldots, e_{L^*})^\top \in \mathbb{R}^{(L^*+1) \times d}$ denotes the learnable embedding matrix containing the embedding vector for the $\texttt{[CLS]}$ token, $q > 0$ controls the form of the fused-type penalty, and $L^* = \max_j L_j$. Note that we share the bin embedding matrix across numerical variables, mapping their discretized values into a common representation space and reducing the number of learnable parameters.

The fused-type regularization is applied in both pretraining and fine-tuning. During pretraining, it enforces local

smoothness between adjacent bin embeddings, encouraging smooth, stable representations across neighboring bins. During fine-tuning, it adaptively refines the learned embeddings by adjusting the effective granularity of the tokenized representation to better align with the downstream task.

**Variable embeddings**  Bin embeddings may not fully capture the heterogeneous semantics of individual variables. A common approach to addressing this limitation is to incorporate *variable embeddings*, analogous to positional embeddings in language models. These variable embeddings provide variable-specific information, allowing the model to recognize feature-level distinctions and increase representational diversity across variables.

Let $\boldsymbol{v}_j \in \mathbb{R}^d$ be the learnable variable embedding for the $j$-th variable. We combine the variable embedding with the bin embedding using two strategies:

$$\tilde{\boldsymbol{e}}_{t,j} = \begin{cases} \text{LayerNorm}(\boldsymbol{e}_t + \boldsymbol{v}_j) & \text{(Add)}, \\ \text{LayerNorm}\left(\text{Proj}\left(\text{Concat}(\boldsymbol{e}_t, \boldsymbol{v}_j)\right)\right) & \text{(Concat)}, \end{cases}$$

where $\text{Concat}(\cdot, \cdot)$ stands for the vector concatenation operator and $\text{Proj}(\cdot)$ represents a linear projection mapping $2d$-dimensional vectors to the $d$-dimensional space.

The $\text{Concat}$ formulation can be viewed as a generalization of the embedding concatenation scheme in Zhou et al. (2024). In particular, when the projection layer has no bias, and the output is fed to the Transformer without an intervening LayerNorm, the $\text{Concat}$ method reduces to their formulation. The fused-type regularization applied to bin embeddings can introduce an inductive bias that leads to scale misalignment between bin and variable embeddings. The learnable projection in the $\text{Concat}$ method provides additional flexibility to balance the two components when forming the final token representation.

### 3.3. Masked Self-Supervised Pretraining

Learning conditional dependencies among input variables allows the model to exploit the joint structure of tabular data. By modeling the conditional distribution, the model captures feature interactions and statistical dependencies that are encoded in the learned representation space. This provides more structured embeddings and useful priors for downstream tasks.

To this end, we adopt masked self-supervised pretraining in which a subset of tokens is randomly masked, and the model is trained to recover them from the remaining observed tokens. This procedure encourages the model to approximate the conditional distribution of the masked tokens given their context, promoting contextualized representations that reflect cross-token dependencies.

**Model architecture**  The pretraining architecture of TABULARBERT builds on the BERT encoder architecture with stacked Transformer encoder layers. Specifically, the encoder consists of $N$ Transformer layers with $H$ attention heads. Unlike the original BERT, TABULARBERT is equipped with binning-based tokenization and the tabular token embedding module described earlier.

On the output side, we employ a dual-head prediction layer for masked pretraining: (i) a classifier that predicts the masked token identity according to our tabular tokenization scheme, and (ii) a regressor that reconstructs the corresponding masked original continuous value. Both heads operate on masked positions and are implemented as linear projections, enabling the objectives introduced later to enforce the desired geometric structure in the embedding space more directly.

The overall pretraining pipeline is illustrated in Figure 1. Given a raw tabular instance, we first tokenize it via binning and form tabular token embeddings, which are fed into the encoder to obtain contextualized representations $\boldsymbol{z}_j \in \mathbb{R}^d$:

$$(\boldsymbol{z}_{\texttt{[CLS]}}, \boldsymbol{z}_1, \ldots, \boldsymbol{z}_p) = \text{BERT}_\phi(\tilde{\boldsymbol{e}}_{\texttt{[CLS]}}, \tilde{\boldsymbol{e}}_{t_1,1}, \ldots, \tilde{\boldsymbol{e}}_{t_p,p}),$$

where $\phi$ denotes the parameters of the BERT encoder. The dual-head module is then applied at the masked positions to perform masked self-supervised pretraining.

**Masking scheme**  We adopt the masking scheme of BERT for masked self-supervised pretraining. This strategy includes the $\texttt{[MASK]}$, random-token, and unchanged-token replacement rules.

Given a tokenized tabular instance $\boldsymbol{t} = (t_1, \ldots, t_p)^\top$, we sample a binary mask indicator $m_j \in \{0, 1\}$ for each token position $j$ independently from a Bernoulli distribution with probability $\rho$, where $m_j = 1$ denotes that the token is masked and $m_j = 0$ indicates that it is not masked. We then construct a corrupted instance $\tilde{\boldsymbol{t}} = (\tilde{t}_1, \ldots, \tilde{t}_p)^\top$ as follows:

$$\tilde{t}_j = \begin{cases} t_j, & m_j = 0, \\ u_j, & m_j = 1, \end{cases}$$

where $u_j \in \{\texttt{[MASK]}, r_j, t_j\}$ is drawn according to a categorical distribution with probabilities $(\rho_{\text{mask}}, \rho_{\text{rand}}, \rho_{\text{keep}})$, and $r_j \sim \text{Unif}(\{1, \ldots, L_j\})$.

Random-token replacement ($r_j$) adds feature-level noise, encouraging context-based recovery instead of reliance on $\texttt{[MASK]}$. Leaving a small fraction of masked positions unchanged reduces the mismatch between the pretraining and fine-tuning, since the $\texttt{[MASK]}$ token does not appear during fine-tuning.

**Pretraining objectives**  When input variables are transformed into discrete tokens, a natural approach for learning

their conditional distributions is to use the standard cross-entropy (CE) loss. However, for numerical variables, the resulting contextualized representations should account for the ordinal structure of the underlying continuous values. This is not enforced by CE, and it is also not guaranteed by fused-type regularization applied to the input token embeddings, because ordinal structure may not be preserved after passing through the nonlinear Transformer encoder.

To encourage ordinal structure in contextualized representations, we introduce a squared Wasserstein distance (SWD) loss. Let $\boldsymbol{m} = (m_1, \ldots, m_p)^\top \in \{0,1\}^p$ be a mask vector. We denote the model-predicted conditional probability produced by the classifier head by $\hat{\pi}_j = \Pr(\cdot|\tilde{\boldsymbol{t}}; \Theta)$, where $\Theta$ is the set of model parameters. The SWD loss is defined as

$$\mathcal{L}_{\mathrm{SWD}} = \mathbb{E}_{\boldsymbol{\xi}}\Big[ \sum_{j:m_j=1} \big\| \mathrm{CDF}\,(\boldsymbol{\delta}_{t_j}) - \mathrm{CDF}\,(\hat{\pi}_j) \big\|_2^2 \Big],$$

where $\boldsymbol{\delta}_{t_j} = (\delta_{t_j}(1), \ldots, \delta_{t_j}(L_j))^\top \in \{0,1\}^{L_j}$ denotes the binary vector whose $t_j$-th entry is 1 and all other entries are 0. The expectation is taken over $\boldsymbol{\xi} = (\boldsymbol{m}, \boldsymbol{u})$ with $\boldsymbol{u} = (u_1, \ldots, u_p)^\top$, and $\mathrm{CDF}(\cdot)$ denotes the cumulative distribution function.

In contrast to CE, the SWD loss naturally imposes a distance-aware structure by assigning smaller penalties to predictions that assign higher probabilities to tokens closer to the target token. Importantly, we apply the SWD loss not only to numerical variables but also to categorical variables because SWD-based learning serves as an alternative to the standard CE loss in classification, supported by Proposition 3.1.

**Proposition 3.1** (Upper bound of total variation). *Let $\pi^*, \hat{\pi} \in \Delta^L$ be two discrete probability distributions over a finite ordered support $\{1, 2, \ldots, L\}$, and let $T \sim \pi^*$. Define $\boldsymbol{\delta}_T \in \{0,1\}^L$ as the binary vector whose $T$-th entry is 1 and all other entries are 0. Then*

$$\mathrm{TV}(\pi^*, \hat{\pi}) \leq \sqrt{L}\,\big(\mathbb{E}_T\big[W_2(\boldsymbol{\delta}_T, \hat{\pi})^2\big]\big)^{\frac{1}{2}},$$

*where $\mathrm{TV}(\cdot, \cdot)$ denotes the total variation distance and $W_2(\cdot, \cdot) = \|\mathrm{CDF}(\cdot) - \mathrm{CDF}(\cdot)\|_2$.*

Proposition 3.1 shows that the total variation distance between the true conditional token distribution and the model prediction is upper-bounded by the square root of the expected SWD. This provides a theoretical motivation for using the SWD loss as an alternative to the standard CE loss, also for categorical variables. Moreover, for numerical variables, minimizing the SWD loss encourages the contextualized representations to account for the ordinal structure induced by binning while approximating the underlying conditional token distribution.

Binning-based tokenization inevitably entails information loss from the original numerical variables, as it omits within-bin variability among values. However, such local variability can be informative for capturing fine-grained patterns. To

mitigate this loss, we augment the encoder with an auxiliary regression head that reconstructs the underlying numerical values from their discretized token representations.

Let $I_C$ be the index set of numerical variables. Let $\hat{\boldsymbol{x}} = (\hat{x}_j)_{j \in I_C}^\top = f(\tilde{\boldsymbol{t}}; \Theta)$ denote the regression-head outputs of the full model. We define the mean squared error (MSE) loss as

$$\mathcal{L}_{\mathrm{MSE}} = \mathbb{E}_{\boldsymbol{\xi}}\Big[ \sum_{j \in I_C : m_j = 1} (x_j - \hat{x}_j)^2 \Big].$$

Minimizing MSE encourages the regression head to approximate the conditional expectation of the original values. In other words, the model $f$ learns a denoised reconstruction that captures within-bin variability. Such fine-grained information is encoded in the contextualized embeddings.

We jointly train the model by minimizing a weighted combination of the SWD and MSE losses, i.e.,

$$\mathcal{L} = \mathcal{L}_{\mathrm{SWD}} + \lambda \mathcal{L}_{\mathrm{MSE}},$$

where $\lambda \geq 0$ controls the contribution of the MSE loss. The expectations with respect to $\boldsymbol{\xi}$ are approximated via Monte Carlo sampling.

### 3.4. Fine-Tuning for Downstream Tasks

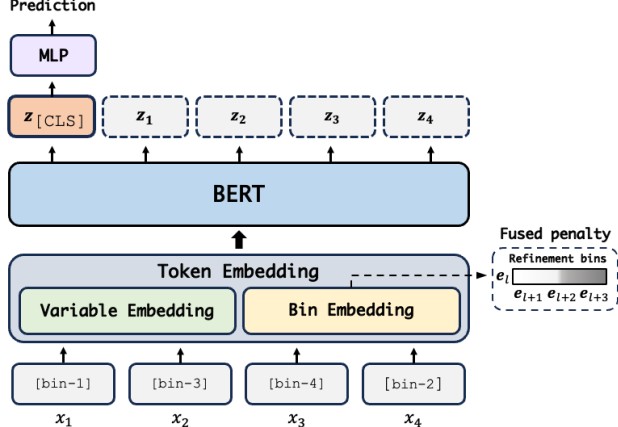

*Figure 2.* Fine-tuning architecture of TABULARBERT. Token embeddings are encoded by a BERT encoder; the [CLS] representation is fed to an MLP head for prediction. The fused-type penalty is retained on bin embeddings during fine-tuning.

For downstream tasks, we replace the pretraining dual-head with a task-specific prediction head on top of the [CLS] representation. We then fine-tune the entire model end-to-end on labeled data so that the bin embeddings can adapt in a task-dependent manner under the fused-type regularization used during fine-tuning. The prediction head is an MLP, trained with CE loss for classification and MSE loss for regression. The fine-tuning architecture of TABULARBERT is shown in Figure 2.

# 4. Experiments

## 4.1. Experimental Setup

This section presents the experimental setup and empirical evaluation of TABULARBERT[1]. We compare our method with tree-based gradient boosting methods and tabular deep learning methods on 12 public tabular datasets, and conduct ablation and representation analyses to assess the contributions of our objectives and regularization. Our experimental study is designed to address the following questions:

- Does the proposed method achieve strong predictive performance on tabular prediction tasks?

- Are the fused-type regularization and pretraining objectives effective for improving downstream performance?

- Do pretraining and fine-tuning yield structured and aligned representation spaces?

- Can the model provide qualitative insights into relationships among input variables?

**Datasets**   We use 12 public tabular datasets, summarized in Table 1. All datasets follow the fixed train/validation/test splits used in Gorishniy et al. (2021; 2022); Yan et al. (2023). Additional dataset details and full names of abbreviations are provided in Appendix B.

*Table 1.* Summary of the 12 public datasets. Acc. indicates accuracy, and RMSE represents root mean squared error.

| Dataset | # samples | # features | # classes | Metric |
|---------|-----------|------------|-----------|--------|
| CH | 10,000 | 10 | 2 | Acc. |
| AD | 48,842 | 14 | 2 | Acc. |
| HI | 98,050 | 28 | 2 | Acc. |
| GE | 9,873 | 32 | 5 | Acc. |
| EY | 10,936 | 26 | 3 | Acc. |
| OT | 61,878 | 93 | 9 | Acc. |
| HE | 65,196 | 27 | 100 | Acc. |
| JA | 83,733 | 54 | 4 | Acc. |
| CA | 20,640 | 8 | – | RMSE |
| HO | 22,784 | 16 | – | RMSE |
| FB | 197,080 | 51 | – | RMSE |
| YE | 515,345 | 90 | – | RMSE |

**Comparison methods**   We compare our TABULARBERT with a diverse set of competitive baselines, including both tree-based gradient boosting methods and deep learning models for tabular data. Specifically, we consider XGBoost (Chen & Guestrin, 2016), LightGBM (Ke et al., 2017), and CatBoost (Prokhorenkova et al., 2018), which are strong and widely used baselines for tabular predic-

tion tasks. We also compare against deep learning approaches, including an MLP, SNN (MLP with SELU activation), ResNet (He et al., 2016), NODE (Popov et al., 2020), DCNv2 (Wang et al., 2021), SAINT (Somepalli et al., 2021), FT-Transformer (Gorishniy et al., 2021), T2G-Former (Yan et al., 2023), and SSL with BinRecon loss (Lee et al., 2024).

**Implementation details**   Each dataset is preprocessed using a quantile transformation that maps each input variable to an approximately uniform distribution. For regression tasks, the target variable is standardized for all methods.

Hyperparameters for all models are selected via Bayesian optimization using the Optuna library (Akiba et al., 2019), optimizing validation accuracy for classification and validation root mean squared error (RMSE) for regression. The search spaces for tree-based methods are based on Bentéjac et al. (2021); Gorishniy et al. (2021). For MLP, SNN, ResNet, NODE, DCNv2, FT-Transformer, and T2G-Former, we follow the implementations and hyperparameter search spaces in Gorishniy et al. (2021); Yan et al. (2023). We use the official SAINT implementation of Somepalli et al. (2021) and define its search space guided by Zabërgja et al. (2024). For SSL with BinRecon (Lee et al., 2024), we report the results from the original work, as the authors' released code does not fully specify all experimental settings required for faithful reproduction. For TABULARBERT, we use AdamW (Loshchilov & Hutter, 2019) with masking probability $\rho = 0.15$ and $(\rho_{\text{mask}}, \rho_{\text{rand}}, \rho_{\text{keep}}) = (0.8, 0.1, 0.1)$. We fix the number of bins for numerical variables to $L_j = 50$ and set the reconstruction loss weight to $\lambda = 1$ during pretraining. Additional details are provided in Appendix C. All models are implemented in `PyTorch`. Experiments are conducted on an NVIDIA RTX 5090 GPU.

## 4.2. Experimental Results

**Performance comparison**   Evaluation is conducted using classification accuracy and regression RMSE, and each experiment is replicated 20 times with different random seeds. For each seed, we select the checkpoint with the best validation score and record its performance on the test set. The average test performance is reported in Table 2.

TABULARBERT achieves strong performance relative to the deep learning baselines on most benchmark datasets. It also remains highly competitive with tree-based methods, outperforming the best tree-based baseline on many datasets. Notably, TABULARBERT yields substantial gains on HI and EY. These results highlight that TABULARBERT can serve as a strong alternative for tabular prediction tasks.

**Effects of the number of bins and fused-type regularization**   Figure 3 shows the evaluation performance of TABULARBERT under varying numbers of bins ($L_j$) and

---

[1]The implementation code is available at https://github.com/bbeomjin/tabularbert.

*Table 2.* Performance comparison across 12 benchmark datasets. Reported values are the average over 20 trials with different random seeds. ↑ (↓) indicates that higher (lower) is better. Boldface highlights the best-performing deep learning method for each dataset. "–" denotes results that are not directly available. Detailed results including the mean and standard deviation are reported in Appendix D.

| Method | Binary classification | | | Multicategory classification | | | | | Regression | | | |
|---|---|---|---|---|---|---|---|---|---|---|---|---|
| | CH ↑ | AD ↑ | HI ↑ | GE ↑ | EY ↑ | OT ↑ | HE ↑ | JA ↑ | CA ↓ | HO ↓ | FB ↓ | YE ↓ |
| XGBoost | 0.862 | 0.874 | 0.724 | 0.683 | 0.718 | 0.827 | 0.374 | 0.723 | 0.434 | 3.191 | 5.435 | 8.874 |
| LightGBM | 0.862 | 0.873 | 0.722 | 0.668 | 0.708 | 0.817 | 0.371 | 0.718 | 0.435 | 3.189 | 5.463 | 8.955 |
| CatBoost | 0.860 | 0.875 | 0.726 | 0.677 | 0.735 | 0.823 | 0.383 | 0.724 | 0.433 | 3.133 | 5.329 | 8.924 |
| MLP | 0.853 | 0.849 | 0.713 | 0.660 | 0.583 | 0.785 | 0.384 | 0.713 | 0.494 | 3.297 | 5.847 | 8.980 |
| SNN | 0.854 | 0.848 | 0.715 | 0.647 | 0.599 | 0.781 | 0.371 | 0.714 | 0.504 | 3.216 | 5.983 | 8.956 |
| ResNet | 0.856 | 0.847 | 0.722 | 0.675 | 0.613 | 0.791 | 0.396 | 0.722 | 0.488 | 3.295 | 5.803 | 9.025 |
| NODE | 0.852 | 0.849 | 0.720 | 0.654 | 0.594 | 0.799 | 0.375 | 0.723 | 0.471 | 3.220 | 5.879 | 8.844 |
| DCNv2 | 0.854 | 0.849 | 0.714 | 0.652 | 0.583 | 0.785 | 0.385 | 0.714 | 0.485 | 3.239 | 5.920 | 9.027 |
| SAINT | 0.859 | 0.857 | 0.723 | 0.715 | 0.707 | 0.809 | 0.387 | 0.720 | 0.460 | 3.265 | 5.694 | 9.206 |
| FT-Transformer | 0.860 | 0.857 | 0.727 | 0.681 | 0.697 | 0.808 | 0.389 | 0.733 | 0.460 | 3.157 | 5.767 | 8.852 |
| T2G-Former | 0.859 | 0.857 | 0.732 | 0.703 | 0.758 | 0.807 | 0.388 | 0.736 | 0.466 | 3.253 | 5.768 | 8.925 |
| SSL (BinRecon) | 0.843 | 0.857 | 0.737 | 0.720 | – | **0.817** | 0.388 | – | 0.464 | **2.989** | – | – |
| TABULARBERT | **0.863** | **0.862** | **0.760** | **0.733** | **0.831** | 0.817 | **0.398** | **0.749** | **0.441** | 3.113 | **5.414** | **8.680** |

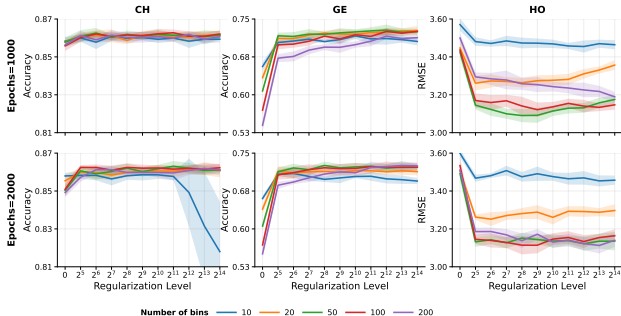

*Figure 3.* Performance across the number of bins, regularization levels, and pretraining epochs. The top row corresponds to 1,000 pretraining epochs and the bottom row to 2,000. Solid lines show the mean performance over 10 replicates, and the shaded regions indicate ±1 standard deviation.

For the largest bin count ($L_j = 200$), performance can remain inferior to smaller bin settings even under strong regularization. One possible explanation is suggested by Proposition 3.1, whose bound scales with $\sqrt{L}$. As $L_j$ increases, a fixed number of pretraining epochs may be insufficient to reduce the discrepancy between the true conditional token distribution and the model prediction. This can be alleviated by increasing the pretraining epochs (see the results for 1,000 and 2,000 pretraining epochs in Figure 3).

*Table 3.* Effect of the SWD objective during pretraining. We compare downstream performance when using SWD and when replacing it with CE.

| | AD ↑ | GE ↑ | EY ↑ | CA ↓ | HO ↓ |
|---|---|---|---|---|---|
| SWD | 0.862 | 0.733 | 0.831 | 0.441 | 3.113 |
| CE | 0.858 | 0.715 | 0.787 | 0.440 | 3.149 |
| SWD − CE | 0.004 | 0.018 | 0.044 | 0.001 | -0.036 |

fused-type regularization levels on the CH, GE, and HO datasets, with all other hyperparameters fixed to their optimized values.

Without regularization, using fewer bins often yields better performance. This observation suggests that using a large number of bins without enforcing proximity across adjacent bins can introduce high variance among bin representations, potentially hindering generalization. As the regularization level increases, performance typically improves, indicating that fused-type regularization plays an important role in stabilizing bin embeddings.

The effect of regularization also depends on the bin count. With a small number of bins, strong regularization can degrade performance, whereas it tends to be beneficial when the number of bins is larger. These trends suggest that finer binning generally requires stronger regularization to control the variability induced by discretization.

**Ablation on the pretraining objectives** To assess the contribution of the SWD objective, we replace it with the CE objective while keeping the reconstruction objective and all other settings fixed. Table 3 reports the downstream performance under the SWD and CE-based pretraining objectives. Using SWD generally improves performance over CE on most datasets, suggesting that encouraging contextualized representations to account for the ordinal geometry of discretized values during pretraining can be beneficial for downstream prediction.

We next isolate the contribution of the auxiliary reconstruction objective by evaluating TABULARBERT with and without this term while keeping all other settings fixed. Table 4 reports the downstream performance when the reconstruction objective is included or omitted during pretraining on several datasets. Adding the reconstruction objective gen-

*Table 4.* Effect of the auxiliary reconstruction objective during pre-training. We report downstream performance of TABULARBERT with (w/ Recon.) and without (w/o Recon.) the reconstruction loss.

|  | AD ↑ | GE ↑ | EY ↑ | CA ↓ | HO ↓ |
|---|---|---|---|---|---|
| w/ Recon. | 0.862 | 0.733 | 0.831 | 0.441 | 3.113 |
| w/o Recon. | 0.860 | 0.716 | 0.796 | 0.440 | 3.151 |
| w/ − w/o | 0.002 | 0.017 | 0.035 | 0.001 | -0.038 |

erally improves performance across datasets. These results suggest that reconstructing masked continuous values helps the model retain within-bin information from the original numerical variables in the learned contextualized representations.

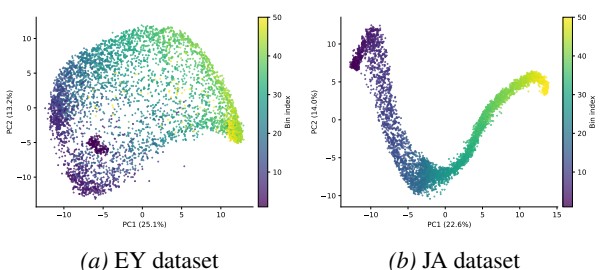

*(a)* EY dataset      *(b)* JA dataset

*Figure 4.* PCA visualizations of contextualized numerical token embeddings after TABULARBERT pretraining on (a) EY and (b) JA.

**Representation space analysis** Figure 4 visualizes the contextualized embeddings of numerical feature tokens after TABULARBERT pretraining by projecting them onto the first two principal components via principal component analysis (PCA). In the resulting PCA space, embeddings associated with nearby bin indices tend to lie closer than those associated with distant bins. This observation suggests that the pretrained representations reflect the ordinal structure of the binned numerical tokens.

We next analyze the fine-tuned contextualized [CLS] embeddings for downstream tasks. Specifically, we perform PCA on these embeddings and compare TABULARBERT with SAINT, which likewise relies on self-supervised pretraining before fine-tuning for downstream prediction.

Figure 5 presents representative examples from classification and regression tasks. On the EY dataset, the TABULARBERT embeddings form more clearly separated class-dependent regions than those of SAINT, which is consistent with its improved classification performance. On the CA dataset, the TABULARBERT embeddings exhibit a smoother target-aligned structure in the PCA space, whereas the SAINT embeddings show a less organized target-dependent pattern. These qualitative patterns suggest that the fine-tuned [CLS] representations learned by TABULARBERT better capture downstream-relevant struc-

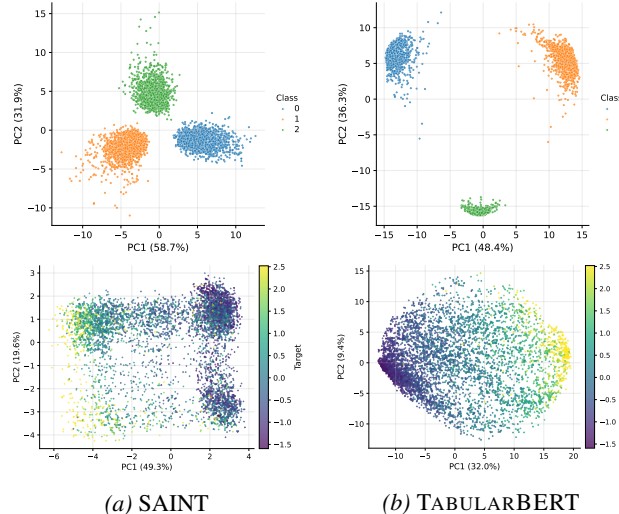

*(a)* SAINT      *(b)* TABULARBERT

*Figure 5.* PCA visualizations of fine-tuned contextualized [CLS] embeddings on EY (top row) and CA (bottom row). The left column corresponds to SAINT and the right column corresponds to TABULARBERT. Colors indicate target values.

ture in these examples.

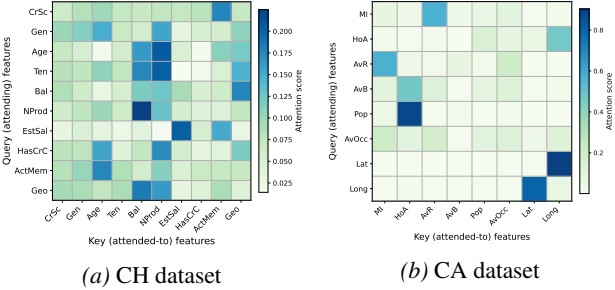

*(a)* CH dataset      *(b)* CA dataset

*Figure 6.* Heatmaps of the averaged first-layer self-attention scores from TABULARBERT for (a) CH and (b) CA.

**Model interpretability** Interpreting attention scores remains debated, partly because model parameters can be non-identifiable. Nevertheless, attention scores offer an intuitive summary of how the model integrates information across feature tokens. Accordingly, to obtain qualitative insights into relationships among input variables, we revisit the CH and CA datasets analyzed in Yan et al. (2023) and examine the self-attention patterns learned by TABULARBERT.

For both datasets, we visualize heatmaps of the first-layer self-attention scores averaged across all instances. The heatmaps are shown in Figure 6, and the variable descriptions and abbreviations are listed in Appendix E. In the CH attention heatmap, strong attention between *NProd* and *Bal* suggests that product holdings may be associated with account balance, potentially reflecting customers' financial engagement and capacity to maintain multiple products. We also observe relatively strong attention from *Age* to *NProd*,

consistent with age-dependent differences in product portfolios that may relate to churn risk.

In the CA attention heatmap, we observe a clear association between *Long* and *Lat*. Notably, *HoA* also exhibits relatively strong attention to *Long*, which is plausibly related to regional development patterns, where coastal areas tend to have older housing age than many inland areas.

## 5. Conclusion

We propose TABULARBERT, a Transformer-based model for tabular data that tokenizes continuous variables via binning and learns numerically structured representations through MLM pretraining and downstream fine-tuning. We introduce fused-type regularization on bin embeddings to encourage smoothness across adjacent bins, a Wasserstein-based objective on contextualized representations to account for ordinal structure, and an auxiliary MSE reconstruction loss to mitigate information loss from discretization.

Our experiments demonstrate that TABULARBERT achieves strong performance relative to deep learning baselines and remains highly competitive with tree-based gradient boosting methods. Empirical results further indicate that the proposed fused-type regularization stabilizes bin embeddings by encouraging local smoothness, improving downstream performance. In addition, the Wasserstein-based objective encourages contextualized representations to account for ordinal structure, while the auxiliary reconstruction objective helps retain within-bin information from discretization. Finally, visualizing self-attention patterns provides qualitative insights into relationships among input variables.

Limitations of TABULARBERT include the increased training complexity introduced by the two-stage pretraining and fine-tuning pipeline, as well as the higher training and inference costs of Transformer-based models compared with lightweight tabular methods. In addition, our framework involves several design choices, including the number of bins and the pretraining schedule. Although the fixed bin number and pretraining schedule used in our experiments worked well as a general configuration, their optimal values may depend on dataset characteristics, and dataset-adaptive selection of these hyperparameters could further improve performance.

## Impact Statement

This paper presents work whose goal is to advance the field of Machine Learning. There are many potential societal consequences of our work, none of which we feel must be specifically highlighted here.

## Acknowledgements

Beomjin Park was supported by the National Research Foundation of Korea (NRF) grant funded by the Korea government (MSIT) (RS-2025-00520073). Seunghwan An was supported by the National Research Foundation of Korea (NRF) grant funded by the Korea government (MSIT) (No. RS-2026-25480090). Sungchul Hong was supported by the National Research Foundation of Korea (NRF) grant funded by the Korea government (MSIT) (RS-2026-25486061). Hosik Choi was supported by the National Research Foundation of Korea (NRF) grant funded by the Korea government (RS-2022-NR068754) and Korea Environment Industry & Technology Institute (KEITI) through Technology Development Project for Safety Management of Household Chemical Products, funded by Korea Ministry of Environment (MOE) (RS-2023-00215309).

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

# A. Proofs

## A.1. Proof of Proposition 3.1

*Proof.* Define $F_\pi(i) = \sum_{l=1}^{i} \pi(l)$. For arbitrary $p, q \in \Delta^L$, let $d_i = F_p(i) - F_q(i)$ with $d_0 = 0$. Since $p(i) - q(i) = d_i - d_{i-1}$, we have

$$
\begin{aligned}
\sum_{i=1}^{L} |p(i) - q(i)| &\leq \sum_{i=1}^{L} (|d_i| + |d_{i-1}|) \\
&= |d_0| + 2\sum_{i=1}^{L-1} |d_i| + |d_L| \\
&\leq 2\sum_{i=1}^{L} |F_p(i) - F_q(i)| \\
&\leq 2\sqrt{L} \left( \sum_{i=1}^{L} (F_p(i) - F_q(i))^2 \right)^{\frac{1}{2}}.
\end{aligned}
$$

On the other hand, for a random variable $T \sim \pi^*$, we have $\mathbb{E}_T[\boldsymbol{\delta}_T] = \pi^*$. Moreover, by Jensen's inequality, we obtain

$$
\begin{aligned}
\mathrm{TV}(\pi^*, \hat{\pi}) &:= \frac{1}{2} \sum_{l=1}^{L} |\pi^*(l) - \hat{\pi}(l)| \\
&= \frac{1}{2} \sum_{l=1}^{L} |\mathbb{E}_T[\delta_T(l)] - \hat{\pi}(l)| \\
&\leq \frac{1}{2} \mathbb{E}_T \left[ \sum_{l=1}^{L} |\delta_T(l) - \hat{\pi}(l)| \right] \\
&= \mathbb{E}_T \left[ \mathrm{TV}(\boldsymbol{\delta}_T, \hat{\pi}) \right].
\end{aligned}
$$

By combining these results, we have

$$
\begin{aligned}
\mathrm{TV}(\pi^*, \hat{\pi}) &\leq \mathbb{E}_T \left[ \mathrm{TV}(\boldsymbol{\delta}_T, \hat{\pi}) \right] \\
&\leq \sqrt{L} \mathbb{E}_T \left[ \left( \sum_{i=1}^{L} (F_{\boldsymbol{\delta}_T}(i) - F_{\hat{\pi}}(i))^2 \right)^{\frac{1}{2}} \right] \\
&\leq \sqrt{L} \left( \mathbb{E}_T \left[ \sum_{i=1}^{L} (F_{\boldsymbol{\delta}_T}(i) - F_{\hat{\pi}}(i))^2 \right] \right)^{\frac{1}{2}},
\end{aligned}
$$

where the last inequality follows from Jensen's inequality. Since $(F_\pi(l))_{l=1}^{L} = \mathrm{CDF}(\pi)$, the proof is complete. $\square$

# B. Additional Dataset Details

The datasets used in our experiments are as follows: Churn Modeling (CH)[2] is a customer-churn prediction dataset from Kaggle; Adult (AD) is a binary income classification task based on census variables (Kohavi, 1996); Higgs Small (HI) is a binary particle-physics classification task distinguishing Higgs events from background (Baldi et al., 2014); Gesture Phase (GE) is a multiclass gesture-phase recognition task derived from kinematic features (Madeo et al., 2013); Eye Movements (EY) is a multiclass dataset constructed from eye-movement features (Salojärvi et al., 2005); Otto Group Product Classification (OT)[3] is a multiclass Kaggle dataset with anonymized numerical features; Helena (HE) and Jannis (JA)

---

[2] https://www.kaggle.com/datasets/shrutimechlearn/churn-modelling
[3] https://www.kaggle.com/c/otto-group-product-classification-challenge

are large-scale tabular classification datasets from the AutoML benchmark suite (Guyon et al., 2019); California Housing (CA) is a regression benchmark for house-value prediction from census attributes (Pace & Barry, 1997); House 16H (HO)[4] is a regression dataset from OpenML; Facebook Comments Volume (FB) is a regression task predicting comment volume from post-related attributes (Singh et al., 2015); and Year (YE) is a regression dataset for predicting the release year of a song from audio-derived features (Bertin-Mahieux et al., 2011). Table 5 summarizes the number of training/validation/test instances, the counts of numerical and categorical variables, and the training batch size for each dataset.

*Table 5.* Dataset statistics used in our experiments

| Dataset | # Train | # Validation | # Test | # Num | # Cat | Task | Batch size |
|---|---|---|---|---|---|---|---|
| Churn Modeling (CH) | 6400 | 1600 | 2000 | 9 | 1 | Binclass | 256 |
| Adult (AD) | 26048 | 6513 | 16281 | 6 | 8 | Binclass | 512 |
| Higgs Small (HI) | 62751 | 15688 | 19610 | 28 | 0 | Binclass | 1024 |
| Gesture Phase (GE) | 6318 | 1580 | 1975 | 32 | 0 | Multiclass | 256 |
| Eye Movements (EY) | 6998 | 1750 | 2188 | 26 | 0 | Multiclass | 256 |
| Otto Group Product Classification (OT) | 39601 | 9901 | 12376 | 93 | 0 | Multiclass | 512 |
| Helena (HE) | 41724 | 10432 | 13040 | 27 | 0 | Multiclass | 512 |
| Jannis (JA) | 53588 | 13398 | 16747 | 54 | 0 | Multiclass | 1024 |
| California Housing (CA) | 13209 | 3303 | 4128 | 8 | 0 | Regression | 512 |
| House 16H (HO) | 14581 | 3646 | 4557 | 16 | 0 | Regression | 512 |
| Facebook Comments Volume (FB) | 157638 | 19722 | 19720 | 50 | 1 | Regression | 1024 |
| Year (YE) | 370972 | 92743 | 51630 | 90 | 0 | Regression | 1024 |

## C. Training and Hyperparameter Optimization Details

### C.1. Tree-Based Gradient Boosting Methods

For the hyperparameter search spaces of XGBoost and CatBoost, we follow Gorishniy et al. (2021). For LightGBM, we adopt a search space analogous to XGBoost, guided by the comparative study of Bentéjac et al. (2021); the full specification is provided in Table 6. We use the GOSS strategy for data sampling in LightGBM.

*Table 6.* LightGBM hyperparameter space

| Parameter | Distribution |
|---|---|
| Max depth | UniformInt$[3, 10]$ |
| Learning rate | LogUniform$[1e\text{-}5, 1]$ |
| Min sum hessian in leaf | LogUniform$[1e\text{-}8, 1e5]$ |
| Feature fraction | Uniform$[0.5, 1]$ |
| Feature fraction by node | Uniform$[0.5, 1]$ |
| Lambda $\ell_1$ | LogUniform$[1e\text{-}8, 1e2]$ |
| Lambda $\ell_2$ | LogUniform$[1e\text{-}8, 1e2]$ |
| Top rate | Uniform$[0.2, 0.7]$ |
| Other rate | Uniform$[0.05, 0.3]$ |
| # iterations | 100 |

### C.2. Baseline Deep Learning Methods

For deep learning baselines, we reuse the implementations and the corresponding hyperparameter search spaces of MLP, SNN, ResNet, NODE, DCNv2, FT-Transformer, and T2G-Former from Yan et al. (2023), and we use the official SAINT implementation of Somepalli et al. (2021). For SAINT, we define the search space guided by Zabërgja et al. (2024); the detailed specification is provided in Table 7.

---

[4]https://www.openml.org/d/574

Hyperparameters are optimized using Optuna (Akiba et al., 2019). All deep learning baselines use the same early-stopping patience of 100, except on the GE dataset. Because the GE dataset exhibits a double-descent phenomenon (Nakkiran et al., 2021), we disable early stopping on GE and train the models for the maximum number of epochs. For baselines other than SAINT, we set the maximum number of epochs to 3,000. For SAINT, the number of pretraining epochs is tuned over [10, 1000], and we set the maximum number of fine-tuning epochs to 2,000. We also include the self-supervised pretraining baseline of Lee et al. (2024); however, the authors' official implementation does not fully specify or provide all experimental settings used in the paper, making faithful reproduction difficult. We therefore report the results as presented in Lee et al. (2024) for this baseline.

Table 7. SAINT hyperparameter space. Here (A) = {CH, AD, HI, GE, EY, OT, HE, JA, CA, HO}, (B) = {FB, YE}.

| Parameter | Distribution |
|---|---|
| Transformer depth | (A, B) UniformInt[1, 4] |
| Embedding size | (A, B) {4, 8, 16, 32, 64} |
| Attention heads | (A, B) {1, 2, 4, 8} |
| Attention dropout | (A, B) Uniform[0, 1] |
| Feed-forward dropout | (A, B) Uniform[0, 1] |
| Learning rate | (A, B) LogUniform[1e-6, 2e-4] |
| Weight decay | (A, B) LogUniform[1e-8, 1e-2] |
| Pretraining epochs | (A, B) UniformInt[10, 1000] |
| # iterations | (A) 100, (B) 50 |

## C.3. TABULARBERT

Our TABULARBERT is trained with the AdamW optimizer (Loshchilov & Hutter, 2019). We set the masking probability to $\rho = 0.15$ and use $(\rho_{\text{mask}}, \rho_{\text{rand}}, \rho_{\text{keep}}) = (0.8, 0.1, 0.1)$. For tokenization, the number of bins for numerical variables is fixed to $L_j = 50$, whereas categorical variables are tokenized using their original number of categories.

During pretraining, we use a fixed learning rate of $2 \times 10^{-4}$ and train for 1,000 epochs to ensure sufficient convergence. We apply the fused-type regularization with the squared $\ell_2$ penalty (i.e., $\|\cdot\|_2^2$) in pretraining, as its primary role is to reduce variation and encourage proximity between embeddings of adjacent bins. The parameter $\lambda$ that controls the contribution of the MSE loss is set to 1.

During fine-tuning, we train for up to 2,000 epochs and use the same early-stopping strategy as the deep learning baselines. All remaining hyperparameters are tuned using Optuna, and the search space is provided in Table 8. To reduce the complexity of hyperparameter optimization, we do not tune the fused regularization weight and weight decay separately for pretraining and fine-tuning; instead, both are tuned once and shared across the two stages.

Table 8. TABULARBERT hyperparameter space. Here (A) = {CH, AD, HI, GE, EY, OT, HE, JA, CA, HO}, (B) = {FB, YE}.

| Parameter | Distribution |
|---|---|
| Transformer layers | (A, B) UniformInt[1, 5] |
| Embedding dimension | (A, B) {32, 64, 128, 256, 512} |
| Attention heads | (A, B) {1, 2, 4, 8} |
| Transformer dropout | (A, B) Uniform[0, 0.5] |
| Embedding fusion | (A, B) {Add, Concat} |
| Fused regularization weight | (A, B) $\{2^4, 2^5, \ldots, 2^{12}\}$ |
| Weight decay | (A, B) LogUniform[1e-8, 1e-2] |
| Learning rate (fine-tuning) | (A, B) LogUniform[1e-6, 2e-4] |
| Head depth (fine-tuning) | (A, B) {0, 1} |
| Head dropout (fine-tuning) | (A, B) Uniform[0, 0.5] |
| Fused regularization $q$ (fine-tuning) | (A, B) {1, 2} |
| # iterations | (A) 100, (B) 50 |

# D. Detailed Experiment Results

*Table 9.* Detailed results for all models on the binary classification tasks. Results are reported as mean ± standard deviation over 20 runs. "–" denotes results that are not directly available.

| Method | Binary classification | | |
|---|---|---|---|
| | CH ↑ | AD ↑ | HI ↑ |
| XGBoost | 0.8622±0.0020 | 0.8741±0.0003 | 0.7240±0.0024 |
| LightGBM | 0.8624±0.0015 | 0.8733±0.0009 | 0.7222±0.0017 |
| CatBoost | 0.8603±0.0008 | 0.8752±0.0006 | 0.7255±0.0010 |
| MLP | 0.8529±0.0046 | 0.8491±0.0010 | 0.7134±0.0013 |
| SNN | 0.8537±0.0037 | 0.8484±0.0014 | 0.7154±0.0017 |
| ResNet | 0.8562±0.0025 | 0.8470±0.0014 | 0.7221±0.0018 |
| NODE | 0.8520±0.0028 | 0.8490±0.0012 | 0.7195±0.0011 |
| DCNv2 | 0.8543±0.0028 | 0.8485±0.0009 | 0.7144±0.0024 |
| SAINT | 0.8586±0.0018 | 0.8566±0.0813 | 0.7226±0.0020 |
| FT-Transformer | 0.8599±0.0016 | 0.8568±0.0015 | 0.7267±0.0017 |
| T2G-Former | 0.8593±0.0032 | 0.8573±0.0011 | 0.7318±0.0016 |
| SSL (BinRecon) | 0.8430± – | 0.8570± – | 0.7370±0.0010 |
| **TabularBERT** | **0.8627±0.0019** | **0.8619±0.0011** | **0.7602±0.0023** |

*Table 10.* Detailed results for all models on the multicategory classification tasks. Results are reported as mean ± standard deviation over 20 runs. "–" denotes results that are not directly available.

| Method | Multicategory classification | | | | |
|---|---|---|---|---|---|
| | GE ↑ | EY ↑ | OT ↑ | HE ↑ | JA ↑ |
| XGBoost | 0.6830±0.0038 | 0.7182±0.0083 | 0.8267±0.0020 | 0.3737±0.0016 | 0.7227±0.0015 |
| LightGBM | 0.6677±0.0112 | 0.7077±0.0056 | 0.8168±0.0028 | 0.3714±0.0016 | 0.7176±0.0012 |
| CatBoost | 0.6766±0.0064 | 0.7349±0.0048 | 0.8227±0.0010 | 0.3826±0.0013 | 0.7242±0.0013 |
| MLP | 0.6598±0.0067 | 0.5827±0.0072 | 0.7846±0.0019 | 0.3842±0.0013 | 0.7132±0.0019 |
| SNN | 0.6468±0.0058 | 0.5994±0.0107 | 0.7806±0.0023 | 0.3714±0.0022 | 0.7137±0.0017 |
| ResNet | 0.6748±0.0062 | 0.6127±0.0102 | 0.7912±0.0013 | 0.3963±0.0018 | 0.7216±0.0018 |
| NODE | 0.6535±0.0063 | 0.5940±0.0085 | 0.7985±0.0016 | 0.3747±0.0025 | 0.7230±0.0016 |
| DCNv2 | 0.6520±0.0080 | 0.5828±0.0072 | 0.7850±0.0031 | 0.3853±0.0014 | 0.7136±0.0014 |
| SAINT | 0.7147±0.0058 | 0.7073±0.0102 | 0.8088±0.0022 | 0.3872±0.0012 | 0.7196±0.0021 |
| FT-Transformer | 0.6806±0.0059 | 0.6973±0.0150 | 0.8082±0.0034 | 0.3886±0.0026 | 0.7332±0.0019 |
| T2G-Former | 0.7034±0.0075 | 0.7575±0.0198 | 0.8072±0.0027 | 0.3880±0.0023 | 0.7361±0.0020 |
| SSL (BinRecon) | 0.7200±0.0020 | – | 0.8170± – | 0.3880±0.0010 | – |
| **TabularBERT** | **0.7332±0.0059** | **0.8311±0.0260** | 0.8173±0.0029 | **0.3979±0.0021** | **0.7489±0.0014** |

*Table 11.* Detailed results for all models on the regression tasks. Results are reported as mean $\pm$ standard deviation over 20 runs. "–" denotes results that are not directly available.

| Method | Regression | | | |
| --- | --- | --- | --- | --- |
| | CA ↓ | HO ↓ | FB ↓ | YE ↓ |
| XGBoost | 0.4335±0.0015 | 3.1909±0.0072 | 5.4346±0.0480 | 8.8739±0.0046 |
| LightGBM | 0.4354±0.0011 | 3.1891±0.0091 | 5.4625±0.0307 | 8.9547±0.0080 |
| CatBoost | 0.4333±0.0025 | 3.1327±0.0140 | 5.3287±0.0199 | 8.9238±0.0116 |
| MLP | 0.4944±0.0027 | 3.2968±0.0435 | 5.8467±0.0438 | 8.9804±0.0364 |
| SNN | 0.5042±0.0086 | 3.2163±0.0167 | 5.9833±0.0444 | 8.9557±0.0278 |
| ResNet | 0.4878±0.0024 | 3.2954±0.0181 | 5.8027±0.0769 | 9.0247±0.0205 |
| NODE | 0.4706±0.0009 | 3.2199±0.0092 | 5.8792±0.0487 | 8.8443±0.0242 |
| DCNv2 | 0.4852±0.0036 | 3.2393±0.0146 | 5.9198±0.0444 | 9.0271±0.0697 |
| SAINT | 0.4603±0.0012 | 3.2649±0.0348 | 5.6936±0.0434 | 9.2055±0.0256 |
| FT-Transformer | 0.4602±0.0034 | 3.1565±0.0352 | 5.7666±0.0694 | 8.8521±0.0354 |
| T2G-Former | 0.4663±0.0067 | 3.2532±0.0347 | 5.7675±0.0612 | 8.9247±0.0464 |
| SSL (BinRecon) | 0.4640±0.0010 | **2.9890±0.0150** | – | – |
| TabularBERT | **0.4413±0.0019** | 3.1133±0.0188 | **5.4137±0.0937** | **8.6801±0.0343** |

# E. Dataset Variable Descriptions

*Table 12.* Variable descriptions for the CH and CA datasets (adapted from Yan et al. (2023)).

| Dataset | Variable | Abbrev. | Description |
| --- | --- | --- | --- |
| CH | CreditScore | *CrSc* | Credit score of the customer. |
| | Gender | *Gen* | Gender of the customer. |
| | Age | *Age* | Age of the customer. |
| | Tenure | *Ten* | Number of years for which the customer has been with the bank. |
| | Balance | *Bal* | Bank balance of the customer. |
| | NumOfProducts | *NProd* | Number of bank products the customer is utilizing. |
| | EstimatedSalary | *EstSal* | Estimated salary of the customer. |
| | HasCrCard | *HasCrC* | Whether the customer has a credit card. |
| | IsActiveMember | *ActMem* | Whether the customer is an active member of the bank. |
| | Geography | *Geo* | Customer's country of residence. |
| CA | MedInc | *MI* | Median income within a block group. |
| | HouseAge | *HoA* | Median house age within a block group. |
| | AveRooms | *AvR* | Average number of rooms per household. |
| | AveBedrms | *AvB* | Average number of bedrooms per household. |
| | Population | *Pop* | Total number of people residing within a block group. |
| | AveOccup | *AvOcc* | Average number of household members within a block group. |
| | Latitude | *Lat* | Block group latitude. |
| | Longitude | *Long* | Block group longitude. |

