# OpenReview forum: "TabularBERT: Binning-Based Self-Supervised Learning for Tabular Representation"
_ICML.cc/2026/Conference — ICML 2026 regular_

### Official Review · Reviewer_rAzt · 2026-03-05

**Soundness:** 2
**Presentation:** 3
**Significance:** 2
**Originality:** 2
**Overall Recommendation:** 3
**Confidence:** 4

**Summary:**

This work proposes a binning-based embedding strategy to obtain the numerical embeddings. A fused-type regularization is proposed to ameliorate the sharp cut in the binning. The squared Wasserstein distance is used in the pretraining process to impose the ordinal relationship on contextualized representations.

**Compliance With Llm Reviewing Policy:**

Affirmed.

**Final Justification:**

I decided to maintain my score. I think the innovation is limited. The self-supervised pre-training has been proposed in tabular representation learning (e.g. [1]). Besides, the baselines are not strong baselines. For example, XGBoost is not a strong tabular learner, but it shows the best performance on several datasets.

[1] Yin, Pengcheng, et al. "TaBERT: Pretraining for joint understanding of textual and tabular data." Proceedings of the 58th annual meeting of the association for computational linguistics. 2020.

**Key Questions For Authors:**

1. On the bottom of page 3, there is an equation for two combination strategies. What is the relationship between the Add and Concat strategies? Are these two strategies applied in a sequential order or at the same time, only one of the combination strategies is applied?

2. How are the model weights of BERT backbone initialized? Are pretrained weights of the BERT model used to initialize the model weight?

**Limitations:**

The proposed method in this work should be able to be applied to any encoder-only models as it proposes an embedding strategy for numerical values. As the name of the proposed method suggests, this work is currently limited to the classic BERT model.

**Strengths And Weaknesses:**

## Strength:

1. Paper is clearly written and has good readability.
2. Using the Squared Wasserstein distance loss to impose an ordinal relationship is innovative.

## Weakness:

### Major Weakness:

1. Questionable claim.

The claim "In contrast to categorical variables, numerical variables are not inherently discretized and thus cannot be directly represented as tokens." is questionable. Numbers can be tokenized into tokens, which is exactly how LLM processes numerical values. The claim requires explanations.

When discussing self-supervised pretraining, "a natural approach for learning their conditional distributions is to use the standard cross entropy (CE) loss." The standard CE requires labels to compute loss, which is contradictory to self-supervised pretraining. Please explain this discrepancy.

2. Innovation is limited.

This work proposes the binning-based tokenization using quantiles. Bin indices are converted to bin embeddings through an embedding layer. A similar idea is proposed in [1]. Self-supervised pre-training is commonly used in training transformer-based models in the tabular data domain [2-4]. Please include those works in Section Related Work and discuss the differences between TabularBERT and them.

3. Baselines are outdated.

The baselines introduced in this work are not SOAT tabular learning baselines. T2G-Former and FT-Transformer are baselines that have the best performance (some results in SSL (BinRecon) are not available, so not included here). They are not new SOTA baselines. Please consider including more recent SOTA baselines. The authors should at least compare the performance of TabularBERT with [1,3,4,5,6].

4. PCA analysis on [CLS] embeddings does not show the advantage of TabularBERT.

Figure 5 shows the PCA visualization on [CLS] embeddings for SAINT and TabularBERT. The top row indicates a similar performance in terms of using decision boundaries to delineate different classes. On the bottom row, SAINT seems to have less overlapping between different class instances compared to TabularBERT.

5. Incomplete results on PCA visualization.

PCA visualization of contextualized numerical token embeddings is helpful to examine the closeness of different bin embeddings. However, only two datasets of multicategory classification are shown. There is no complete visualization for all datasets, which renders Figure 4 potentially vulnerable to a cherry-picking problem. The authors should at least show the PCA visualization for datasets in the binary classification category and the regression category.

6. Result comparison is misleading.

In Table 2, the best performance is marked in bold font. However, the usage of bold font is misleading. For example, XGBoost shows better performance on AD, OT and CA datasets. The authors mark the performance of TabularBERT in bold font, which can mislead readers into believing TabularBERT has better performance on those datasets.

### Minor Weakness:

1. On the bottom Page 5, the equation only cover the regression task. The equation covering binary classification and multiclass classification is missing.

[1] Yan, Jiahuan, et al. "Making pre-trained language models great on tabular prediction." arXiv preprint arXiv:2403.01841 (2024).
[2] Yin, Pengcheng, et al. "TaBERT: Pretraining for joint understanding of textual and tabular data." Proceedings of the 58th annual meeting of the association for computational linguistics. 2020.
[3] Wang, Zifeng, and Jimeng Sun. "Transtab: Learning transferable tabular transformers across tables." Advances in Neural Information Processing Systems 35 (2022): 2902-2915.
[4] Zhu, Bingzhao, et al. "Xtab: Cross-table pretraining for tabular transformers." arXiv preprint arXiv:2305.06090 (2023).
[5] Thielmann, Anton Frederik, et al. "Mambular: A sequential model for tabular deep learning." arXiv preprint arXiv:2408.06291 (2024).
[6] Gorishniy, Yury, Akim Kotelnikov, and Artem Babenko. "Tabm: Advancing tabular deep learning with parameter-efficient ensembling." arXiv preprint arXiv:2410.24210 (2024).

---

> ### Author Rebuttal · Authors · 2026-03-29
>
> ## Question 1
> We thank the reviewer for pointing out this ambiguity.
> `Add` and `Concat` are two alternative fusion strategies, and only one is selected in each run during hyperparameter tuning. `Add` combines the two embeddings directly, whereas `Concat` concatenates them and then projects them back to the hidden dimension. The motivation for `Concat` is that fused regularization on bin embeddings can cause scale misalignment between bin and variable embeddings, and the learnable projection provides additional flexibility to balance them.
>
> ## Question 2
> We thank the reviewer for this question. TabularBERT does not use pretrained NLP BERT weights. We adopt only the BERT-style encoder architecture, and all model parameters are initialized randomly and learned through our masked self-supervised pretraining stage, followed by downstream fine-tuning. This is because our input representation differs from standard BERT, using tabular tokens and variable/bin embeddings rather than text tokens.
>
> ## Major Weakness 1
> We thank the reviewer for these careful comments.
> For the first point, our claim was not that numerical values cannot be tokenized at all, but that, unlike categorical variables, they do not come with a natural finite token vocabulary whose identity directly preserves numerical proximity and order. This is why, in our setting, numerical variables require an additional tokenization step, such as binning.
>
> For the second point, we understand the reviewer’s concern because CE is often introduced in supervised settings.
> However, in masked self-supervised pretraining, the target token at a masked position is generated from the input itself, so CE is computed between the model’s predicted conditional token distribution and the observed masked token identity.
> Our point was that CE is natural for masked token prediction, but does not explicitly preserve the ordinal structure of discretized numerical variables, which motivates SWD.
>
> ## Major Weakness 2
> We thank the reviewer for this helpful comment. We agree that these related works should be discussed more explicitly. Our novelty claim is not that binning, embedding regularization, or self-supervised pretraining are introduced for the first time. Rather, the key distinction of TabularBERT lies in the role these components play. In particular, our approach explicitly distinguishes between (i) embedding-level smoothness, enforced via fused regularization over adjacent bin embeddings, and (ii) ordinal structure preservation in contextualized representations, enforced via a Wasserstein-based objective over predicted conditional distributions. Due to the rebuttal space limit, we respectfully refer the reviewer to our response to **Reviewer uSMg** for the detailed discussion.
>
> ## Major Weakness 3
> We thank the reviewer for this important comment. We agree that the omission of more recent strong baselines is a limitation of the current submission. To partially address this concern, we additionally compared TabularBERT with strong recent baselines such as TabR, TabPFN, and TP-BERTa. Due to the rebuttal space limit, we respectfully refer the reviewer to our response to **Reviewer uSMg** for the detailed results and discussion.
>
> ## Major Weakness 4
> We thank the reviewer for this comment. Figure 5 was intended as a qualitative illustration rather than a standalone claim. To support it quantitatively, we also reported the silhouette scores, and TabularBERT achieved higher values than SAINT. This supports our claim that TabularBERT yields more class-separable representations, and stronger class separation is generally more favorable for downstream generalization.
>
> ## Major Weakness 5
> We agree that Figure 4 shows only selected datasets and is not meant to be exhaustive.
> Its purpose was to provide an illustrative example of how SWD can induce ordinal structure in contextualized numerical token embeddings, rather than a complete visualization across all datasets.
> We also note that the same qualitative pattern was observed in our analysis of binary classification and regression datasets.
> We agree that including such additional examples would further strengthen the presentation.
>
> ## Major Weakness 6
> We thank the reviewer for pointing this out.
> As noted in the caption of Table 2, the boldface was intended to highlight the best-performing deep learning method, not the overall best result across all models. That said, we agree this may still be misread, and we will revise the formatting to make the distinction clearer.
>
> ## Minor Weakness 1
> We thank the reviewer for this comment. We would like to clarify that the equation at the bottom of page 5 is not the downstream loss for regression, but the auxiliary MSE reconstruction objective used during masked self-supervised pretraining for numerical variables. Since this term is introduced only to recover masked numerical values, it does not have binary- or multiclass-classification counterparts in that section.

---

> > ### Author Rebuttal · Reviewer_rAzt · 2026-03-31
> >
> > I thank the authors to address part of my concerns.
> >
> > 1. The authors claim "We agree that Figure 4 shows only selected datasets and is not meant to be exhaustive. Its purpose was to provide an illustrative example of how SWD can induce ordinal structure in contextualized numerical token embeddings, rather than a complete visualization across all datasets.". However, the main manuscript claims "The
> > PCA visualizations in Figure 5 indicate that TABULAR-BERT produces a more class-separable embedding space
> > than SAINT". With only selecting two datasets, it is not convincing that TABULAR-BERT can achieve better class separation.
> >
> > 2. For the performance comparison with recent baselines, without reporting computational costs such as the number of trainable parameters and FLOPs, it is difficult to have a comrehensive evaluation.

---

> > > ### Author Response · Authors · 2026-04-02
> > >
> > > ## Response to Question 1
> > > We thank the reviewer for the follow-up comment. We would like to clarify that Figure 4 and Figure 5 serve different purposes. Figure 4 concerns the pretraining stage and was included to illustrate how the SWD objective can induce ordinal structure in contextualized numerical token embeddings. By contrast, Figure 5 concerns the fine-tuned [CLS] embeddings and was intended as a qualitative analysis tool for understanding the learned downstream representation, rather than as a standalone proof based only on a 2D projection.
> > >
> > > We also agree that showing only two datasets in Figure 5 may leave room for concern about cherry-picking. To address this, we additionally generated PCA visualizations of the fine-tuned [CLS] embeddings for all datasets, and the full set can be found at the following link (https://github.com/Anonymous-Author4/Additional-Figures/blob/main/contextualized_embedding_representation.pdf).
> > >
> > > More specifically, our point is not only that some clusters appear farther apart in 2D, but that the learned embeddings of TabularBERT often exhibit a more structured low-dimensional organization after fine-tuning. This can be observed, for example, on JA and also in the additional AD visualization. While a 2D PCA plot cannot fully represent the whole embedding space, such a structure is still informative, since better-organized representations are generally more favorable for downstream prediction. We would also like to emphasize that these 2D PCA visualizations of the fine-tuned [CLS] embeddings were included as a qualitative interpretation of why TabularBERT showed relatively stronger downstream performance than SAINT, rather than as a standalone quantitative comparison.
> > >
> > > ## Response to Question 2
> > > We thank the reviewer for this important comment. We agree that computational cost is necessary for a more comprehensive comparison with recent baselines. To address this point, we examined the number of trainable parameters and inference FLOPs of representative deep baselines and TabularBERT under the best hyperparameter configurations selected in our experiments. For example, on EY, TabularBERT uses 141.97M FLOPs, compared with 11.77M for FT-Transformer and 11.79M for T2G-Former. On HI, TabularBERT uses 473.01M FLOPs, compared with 3.82M and 3.49M, respectively. For a more detailed comparison of parameter counts and FLOPs across datasets and baselines, we respectfully refer the reviewer to our response to **Reviewer Ajde**.
> > >
> > > We agree that TabularBERT is not intended as a lightweight alternative to smaller tabular models. Rather, it is designed as a representation-learning framework that explicitly models the numerical structure of tabular data through discretized tokenization, conditional dependency learning, and structure-preserving objectives. From this perspective, we believe it is important to consider not only the additional computation, but also whether it leads to meaningful improvements in learned representations and downstream performance.
> > >
> > > We sincerely thank the reviewer again for taking the time to read our paper carefully and for providing constructive and thoughtful comments, which helped us clarify and improve the presentation of our work.

---

### Official Review · Reviewer_Ajde · 2026-03-09

**Soundness:** 3
**Presentation:** 4
**Significance:** 3
**Originality:** 3
**Overall Recommendation:** 5
**Confidence:** 4

**Summary:**

The authors propose TABULARBERT, which is a Transformer-based model. According to their description, it can discretize numerical variables via binning-based tokenization and learns representations that preserve numerical proximity and ordinal information while capturing conditional dependencies among variables through masked self-supervised pretraining.

**Compliance With Llm Reviewing Policy:**

Affirmed.

**Key Questions For Authors:**

1. If transformer architecture has significantly higher computational consumption, the percentage of flops consumption compared to other deep learning methods like TabNet and TabFormer shall be listed.

2. As Bert is quite a big baseline model, how can we be sure the performance elevation is not originating from a larger parameter amount? Maybe a discussion will help us understand the necessity and effect.

**Limitations:**

We acknowledge that this paper is highly novel and original. According to the aforementioned questions, there are still questions to be answered. Meanwhile, the effect os $L$ shall be researched further. As the content of this paper reached the page limits of 8, these limitations can be discussed in future research.

**Strengths And Weaknesses:**

The technical terms and equations are well-described. The writing is clear, and the method is detailed. I take this article as a significant improvement in how we apply language model-based tabular modeling. Though the method is refreshing, the performance elevation is limited. Meanwhile, the effect of $L$ shall be researched more thoroughly. Authors can discuss how and why a small or big $L$ may influence the performance in future research.

---

> ### Author Rebuttal · Authors · 2026-03-29
>
> ## Questions 1 and 2
> We thank the reviewer for these important questions.
> To clarify both computational cost and the role of model size, we additionally report the number of parameters and inference FLOPs of representative deep baselines and TabularBERT in the table below.
> We note, however, that both parameter count and FLOPs depend on the selected hyperparameter configuration.
> The numbers reported here are therefore based on the best configurations chosen in our experiments, and should be interpreted as empirical comparisons under our tuning protocol rather than as immutable properties of the model classes.
> In particular, since we used broad hyperparameter search spaces, FT-Transformer and T2G-Former could also have selected substantially larger configurations if those had been favored by validation performance.
>
> | Dataset | Method | Score | # of Params | FLOPs |
> |---|---|---:|---:|---:|
> | **EY ↑** | FT-Transformer | 0.697 | 583,585 | 11.77 M |
> |  | SAINT | 0.707 | 31,049,499 | 28.86 M |
> |  | T2G-Former | 0.758 | 503,190 | 11.79 M |
> |  | TabularBERT | 0.831 | 5,695,307 | 141.97 M |
> | **HI ↑** | FT-Transformer | 0.727 | 208,729 | 3.82 M |
> |  | SAINT | 0.723 | 308,150 | 0.30 M |
> |  | T2G-Former | 0.732 | 132,048 | 3.49 M |
> |  | TabularBERT | 0.760 | 18,390,962 | 473.01 M |
> | **GE ↑** | FT-Transformer | 0.681 | 771,901 | 15.66 M |
> |  | SAINT | 0.715 | 187,197,219 | 174.17 M |
> |  | T2G-Former | 0.703 | 522,388 | 14.18 M |
> |  | TabularBERT | 0.733 | 4,528,005 | 134.29 M |
> | **CA ↓** | FT-Transformer | 0.460 | 637,591 | 1.56 M |
> |  | SAINT | 0.460 | 426,083 | 0.35 M |
> |  | T2G-Former | 0.466 | 94,126 | 0.80 M |
> |  | TabularBERT | 0.441 | 3,083,570 | 36.60 M |
> | **YE ↓** | FT-Transformer | 8.852 | 358,333 | 7.56 M |
> |  | SAINT | 9.206 | 446,918,013 | 416.55 M |
> |  | T2G-Former | 8.925 | 267,011 | 17.33 M |
> |  | TabularBERT | 8.680 | 6,500,681 | 478.27 M |
>
> From a theoretical perspective, TabularBERT remains in the same complexity class as other Transformer-based tabular models, with a dominant encoder cost of order $O(NL^2d + NLd^2)$, where $L$ is the number of feature tokens, $d$ is the embedding dimension, and $N$ is the number of Transformer layers.
> Thus, the main difference is not a fundamentally different asymptotic scaling, but the concrete configuration selected in practice and the additional cost of the pretraining stage.
>
> Empirically, TabularBERT does use more inference FLOPs than FT-Transformer and T2G-Former on these datasets under the selected best configurations. This confirms that the proposed token-based Transformer modeling is not computationally free. At the same time, the cost is not uniformly higher than all deep baselines: for example, SAINT has substantially more parameters than TabularBERT on several datasets, yet TabularBERT still achieves stronger performance.
>
> We agree that TabularBERT is not a lightweight alternative to smaller tabular models. Rather, it is designed as a representation-learning framework that explicitly models the numerical structure of tabular data through discretized tokenization, conditional dependency learning, and structure-preserving objectives. In this sense, the relevant question is not only whether it uses more computation, but whether that additional computation yields a meaningful improvement in learned representations and downstream performance.
>
> This also helps address the second question. The gains cannot be explained simply by a larger parameter count. In several cases, SAINT uses substantially more parameters than TabularBERT but still performs worse. This suggests that the improvement is not merely a model-size effect, but is tied to the proposed modeling design.

---

### Official Review · Reviewer_uSMg · 2026-03-11

**Soundness:** 2
**Presentation:** 3
**Significance:** 3
**Originality:** 1
**Overall Recommendation:** 2
**Confidence:** 5

**Summary:**

This paper focused on tabular prediction task (classification & regression) and designed a binning-based embedding (tokenization) method for numerical variables to adapt tabular data to BERT. Specifically, column-wise quantile transformation is applied to perform uniform discretization for each numerical features and convert them into bin indices, and all numerical bins are represented by shared lookup embeddings, which are additionally regularized to minimize distance between adjacent bin embeddings during training. The final numerical token embeddings are combined with the bin embedding and positional embedding (called variable embedding). After tokenization, a MLM(masked language modeling)-like self-supervised pre-training is performed with the traditional objective CE loss substituted by SWD loss and regression loss, which are applied separately to retain ordinal structure and to recover original values for numerical features. After dataset-specific self-supervised pre-training and fine-tuning, the resulting TabularBERT surpass a series of tabular deep model and tree model baselines on 12 datasets from FT-Transformer benchmark. The visualization analysis of the learned numerical embeddings indicate the tokenization method captured ordinal structure of numerical features.

**Compliance With Llm Reviewing Policy:**

Affirmed.

**Final Justification:**

The rebuttal has fully responded my concerns, and I would like to increase evaluation of significance accordingly.

**Key Questions For Authors:**

See Weaknesses.

**Limitations:**

yes

**Strengths And Weaknesses:**

# Strengths

- Comprehensive experiment results with average score and deviations are given.

- Visualization analysis is provided to show the representation effectiveness of designed numerical tokenization.

- Formulation proof is given to clarify the superiority of SWD loss to CE loss when adapting MLM-like self-supervised pre-training to tabular data.

# Weakness

- Unclear novelty: To my knowledge, it seems TabularBERT shares a similar framework design as another popular language-model-based tabular transformer paper in ICLR 2024, i.e., TP-BERTa[1], with the major deferences listed below:

| paper       | base model | num. binning   method     | bin embedding penalty    | Pre-training                         |
|-------------|------------|---------------------------------|--------------------------|--------------------------------------|
| TP-BERTa    | RoBERTa    | target-aware C4.5 tree          | relative ranking loss    | cross-dataset supervised learning    |
| TabularBERT | BERT       | uniform quantile transformation | L\_p like regularization | dataset-specific self-supervised learning |
|             |            |                                 |                          |                                      |

Therefore, it makes me hard to judge the real novelty of TabularBERT since it strongly resemble TP-BERTa, and some conclusion has been demonstrated in that paper (e.g., bin embedding regularization stabilize representation and improve performance), while each part is substituted by another method, and some substitutions bring other weaknesses.

- Increased computation cost: Compared to other selected non-langage-model dataset-specific baselines, TabularBERT is designed with several magnitude larger model parameter and heavy pre-training, it naturally suffers from large training budget and inference latency in the experiment, leading to unfair comparison to some extent.

- Missing baselines, data diversity and model usability analysis: Recent popular tabular models include, i) ICL-based models like TabPFN[2], TabICL[3], MotherNet[4], ii) supervisedly pre-trained models like TP-BERTa, CM2[5], TransTab[6], XTab[7], iii) other non-pretrained baselines like TabR[8], ExcelFormer[9], RealMLP[10]. More data diversity and model usability analysis is missing to figure out unique cost-effectiveness of TabularBERT compared to these existing models.

### Reference

[1] Making Pre-trained Language Models Great on Tabular Prediction, ICLR 2024.

[2] Accurate predictions on small data with a tabular foundation model, Nature 2025.

[3] TabICL: A Tabular Foundation Model for In-Context Learning on Large Data, ICML 2025.

[4] MotherNet: Fast Training and Inference via Hyper-Network Transformers, ICLR 2025.

[5] Towards Cross-Table Masked Pretraining for Web Data Mining, WWW 2024.

[6] TransTab: Learning Transferable Tabular Transformers Across Tables, NeurIPS 2022.

[7] XTab: Cross-table Pretraining for Tabular Transformers, ICML 2023.

[8] TabR: Unlocking the Power of Retrieval-Augmented Tabular Deep Learning, ICLR 2024.

[9] Can a Deep Learning Model be a Sure Bet for Tabular Prediction? KDD 2024.

[10] Better by Default: Strong Pre-Tuned MLPs and Boosted Trees on Tabular Data, NeurIPS 2024.

---

> ### Author Rebuttal · Authors · 2026-03-28
>
> ## Weakness 1
> We thank the reviewer for pointing out the connection to TP-BERTa.
> We agree that both works belong to the broader family of discretization-based Transformer models for tabular data, and that both use regularization on embeddings.
> Our claim of novelty is not that such regularization is introduced for the first time.
> Rather, the key difference lies in what the regularization is designed to enforce, how it is formulated, and at what stage it remains active.
> In TP-BERTa, target-aware tokenization and ranking-based regularization are used to construct and stabilize magnitude representations for cross-dataset supervised pretraining.
> While such ranking-based regularization can encourage ordinal consistency, it does not explicitly enforce adjacent-bin smoothness.
> By contrast, TabularBERT introduces a fused-type penalty that directly regularizes local variation between neighboring bins, and this remains active both pretraining and fine-tuning.
> As a result, the discretized representation can continue to be refined for the downstream task, rather than serving only as an initialization-level stabilizer.
> We view this as important because even a strong initial binning scheme does not necessarily define the bin geometry that is most suitable for the final nonlinear Transformer model.
>
> A second difference is that embedding-level smoothness alone does not guarantee that the contextualized representations used for prediction preserve ordinal structure after passing through a deep nonlinear encoder.
> This is exactly why TabularBERT does not rely on fused regularization alone.
> In addition to smoothing adjacent bin embeddings, we explicitly impose a Wasserstein-based objective on the predicted conditional distribution over ordered tokens, so that ordinal structure is preserved in the contextualized representations.
> This is the key reason why we view TabularBERT not as a variant of TP-BERTa, but as a framework for preserving ordered conditional structure in contextualized tabular representations.
>
> ## Weakness 2
> We acknowledge that TabularBERT requires a larger training budget than single-stage baselines because it includes a separate self-supervised pretraining stage before fine-tuning. Due to the rebuttal space limit, we respectfully refer the reviewer to our response to **Reviewer Ajde** for the detailed empirical comparison of parameter counts and FLOPs. The main additional overhead comes from pretraining; at inference time, the model uses the fine-tuned Transformer encoder in the same general way as other Transformer-based tabular models.
>
> ## Weakness 3
> We agree that not all relevant recent baselines were included in the current submission, and we appreciate the reviewer for pointing this out.
> To partially address this concern, we additionally included results for three recent strong baselines: TabR, TabPFN, and TP-BERTa.
>
> For these additional comparisons, we used the authors’ official implementations and matched the main experimental settings of TabularBERT whenever applicable. All reported results are based on 20 repeated runs. For TP-BERTa, since it is designed for cross-dataset supervised pretraining and its full training is substantially more time-consuming, we used the authors' provided pretrained backbone and performed fine-tuning only. Even fine-tuning was relatively expensive, and due to the rebuttal time limit, we were not able to obtain results on all datasets.
> The resulting performance is shown below:
>
> | Model | AD ↑ | HI ↑ | GE ↑ | JA ↑ | YE ↓ | KDD ↑ |
> |---|---:|---:|---:|---:|---:|---:|
> | TabularBERT | 0.862 | **0.760** | 0.733 | **0.749** | **8.680** | **0.801** |
> | TabR | **0.870** | 0.725 | 0.744 | 0.741 | 8.982 | 0.779 |
> | TabPFN | 0.860 | 0.719 | **0.784** | 0.723 | 8.948 | 0.794 |
> | TP-BERTa | 0.867 | 0.707 | 0.491 | - | - | 0.790 |
>
> These additional results suggest that TabularBERT remains competitive against strong recent baselines.
> In particular, it performs strongly on HI, JA, and YE.
> We conjecture that these are settings where explicit conditional modeling across variables is especially useful, such as datasets with substantial feature redundancy or heterogeneous/noisy predictors.
> For example, HI contains both detector-level features and high-level derived variables.
>
> To further support this point, we additionally evaluated KDDCUP, which was not part of the original submission.
> We included this dataset as an extra analysis case because it contains heterogeneous and noisy variables with varying relevance to the target, making it a useful setting for probing whether explicit conditional modeling remains beneficial.
> On this dataset as well, TabularBERT outperformed TabR, TabPFN, and TP-BERTa.
>
> At the same time, we do not claim uniform superiority over all recent tabular methods.
> Rather, these additional results are intended to partially address the reviewer’s concern and to show that TabularBERT remains competitive relative to recent strong baselines.

---

> > ### Author Rebuttal · Reviewer_uSMg · 2026-04-02
> >
> > Thanks for the authors providing additional recent baseline comparison on several datasets, which to some extent exhibits the significance of TabularBERT performance over the recent advanced baseline on certain data regime.
> >
> > - As responded in weakness 1, the authors recognized the connections between TP-BERTa and TabularBERT, both are numerical-feature-discretization-based Transformer models for tabular data, and both use regularization to learn effective numerical embeddings. The main difference is that TP-BERTa used target-aware discretization, ranking-based embedding regularization, and cross-table supervised pre-training; while TabularBERT used uniform quantile discretization, neighboring-bin regularization, and self-supervised pre-training respectively. As the authors stated ("We view this as important because ... is most suitable for the final nonlinear Transformer model" & "In addition to smoothing adjacent bin embeddings ... ordinal structure is preserved in the contextualized representations"), **the real distinction of TabularBERT lies in its improved and more suitable numerical embedding and contextual representation method**; therefore, it is not regarded as a variant of TP‑BERTa. I agree that TabularBERT is not same as TP-BERTa, just like BERT is not same as RoBERTa, ALBERT, SpanBERT, **while the level of innovation remains a matter of subjective judgment. It should be noted that some reviewers may not be familiar with prior related studies, and such discretization-based tabular Transformer was not mentioned or compared in the paper**.
> >
> > - As responded in weakness 2, the authors agreed the TabularBERT requires a larger training budget than directly fine-tuning baselines since its separated pre-training and fine-tuning for each dataset, while TabularBERT uses the language model (BERT) as backbone, though during fine-tuning all baselines are Transformers, **TabularBERT still suffers from more larger computation cost in both training and inference (compared to non-language-model baselines like FT-Transformer, TaR), which inevitably impact its practically usability**.
> >
> > - For response in weakness 3, advanced baselines like TabPFN, TP-BERTa are extensively evaluated on hundreds of downstream datasets to demonstrate their general effectiveness, the results evaluated on several datasets may lack statistical significance, while **I agree TabularBERT is able to exhibit significant results on suitable data regime, I think it will be better to analyze the data scope in which TabularBERT excels to provide the audience with better usability guidance (like TabPFN excels in small to medium tabular data, TP-BERTa performs better on categorical feature dominated dataset)**.
> >
> >
> > Overall, the rebuttal has fully responded my concerns, and I would like to increase evaluation of significance accordingly, thank you.

---

> > > ### Author Response · Authors · 2026-04-02
> > >
> > > We sincerely thank the reviewer for taking the time to read our paper and rebuttal so carefully, and for providing such thoughtful, constructive, and helpful feedback. We greatly appreciate the reviewer’s recognition of the distinctions we aimed to clarify, as well as the additional suggestions regarding practical usability and the data regimes in which TabularBERT may be most effective. These comments are very meaningful to us and will be highly valuable in improving both the presentation and positioning of our work.

---

### Official Review · Reviewer_qKKs · 2026-03-11

**Soundness:** 3
**Presentation:** 4
**Significance:** 3
**Originality:** 2
**Overall Recommendation:** 4
**Confidence:** 4

**Summary:**

The paper proposes TabularBERT, a transformer architecture for tabular data that discretizes numerical features using quantile-based binning and learns representations through masked language modeling (MLM) pretraining followed by task-specific fine-tuning. The key contributions are as follows: 1) a fused-type regularization on bin embeddings to enforce local smoothness between adjacent bins, 2) a squared Wasserstein distance (SWD) loss to impose ordinal structure on the representations, and 3) an auxiliary MSE reconstruction loss to mitigate within-bin information loss caused by discretization. The model is evaluated on 12 public tabular datasets across classification and regression tasks and is compared against tree-based methods (XGBoost, LightGBM, CatBoost) and several deep learning baselines (MLP, ResNet, NODE, SAINT, FT-Transformer, T2G-Former, and SSL with BinRecon). The paper also includes ablation studies, representation visualizations, and attention-based interpretability analysis.

**Compliance With Llm Reviewing Policy:**

Affirmed.

**Final Justification:**

Overall, this is a solid paper with moderate novelty (introduces new ideas, but several closely related papers exists). Experiments are generally solid but some issues exists, which could not completely be resolved by the rebuttal.

**Key Questions For Authors:**

1. Can you provide a more detailed comparison with Lee et al. (2024) that clearly delineates which architectural and objective function choices are inherited from their work versus genuinely novel to TabularBERT?

2.  Have you considered per-variable bin embedding matrices instead of the shared one? What performance impact would this have, and how does it interact with the fused regularization?

3. What is the computational overhead of TabularBERT's pretraining + fine-tuning pipeline relative to direct supervised training of FT-Transformer or T2G-Former? Please provide wall-clock times or FLOPs comparisons.

4. Why are the results for SSL (BinRecon) missing on EY, JA, FB, and YE? Could you re-implement Lee et al. (2024) under your experimental protocol to enable a fairer comparison? Also, could you explain why you did not experiment with methods such as TabPFN, ModernNCA, and TabR, given that TabR shows superior performance compared to your reported results?

5.  Can you provide an ablation isolating the contribution of SWD loss versus standard cross-entropy? The current ablation (Table 3) only isolates the reconstruction objective, but does not separately evaluate the effect of replacing CE with SWD.

6.  In the released code, the _cat2int method in the discretization module builds category-to-integer mappings using Python's set(), whose iteration order is not guaranteed by the language specification. Since this mapping is rebuilt on each call to discretize() rather than being persisted during fit(), could you confirm that categorical features receive consistent integer codes across training, validation, and test splits? We note that in the provided training script, categories are pre-encoded to sorted integers upstream, which likely makes this safe in practice, but the discretizer itself does not enforce this consistency. For comparison, Lee et al. (2024) handled this by using sorted unique values from training data applied deterministically to all splits.

**Limitations:**

The authors discussed the limitations of the increased complexity and higher training and inference costs caused by their two-stage pretraining and fine-tuning pipeline, although they did not report any corresponding metrics.

**Strengths And Weaknesses:**

Strengths:

1. The paper identifies a challenge—preserving ordinal structure when discretizing numerical features for tokenization—and addresses it through input-level regularization (fused-type penalty), output-level supervision (SWD loss), and value reconstruction (MSE loss). The combination is well motivated, and each component addresses a distinct aspect of information loss caused by binning.

2. A solid experimental protocol is used, including 20 random seeds per experiment, hyperparameter optimization via Optuna, and a detailed appendix with full results. TabularBERT shows strong empirical results, particularly on the HI, EY, and YE datasets. The model also competes favorably with tree-based methods, which is meaningful for deep tabular learning.

3. Fig. 3 clearly demonstrates the interplay between bin count and regularization strength, and the PCA visualizations provide useful qualitative and quantitative evidence that the learned representations preserve ordinal structure.

Weaknesses:

1. The novelty is partial rather than fundamental. The core idea of using binning as a pretext task for self-supervised tabular learning was proposed by Lee et al. (2024), who introduced both the BinRecon and BinXent losses and demonstrated that predicting bin indices rather than raw values yields superior tabular representations. Lee et al. also used quantile-based binning, masked input transformations, and BERT-style architectures (via FT-Transformer). The paper would benefit from a much more explicit and detailed comparison clarifying what is truly new versus inherited from Lee et al. (2024).

2. Incomplete and potentially unfair baseline comparison: The authors report results for SSL with BinRecon (Lee et al., 2024) directly from the original paper, noting that "the authors' released code does not fully specify all experimental settings required for faithful reproduction." This makes direct comparison problematic, as discrepancies could arise from differences in hyperparameter tuning budgets, preprocessing, or evaluation protocols. Several entries for SSL (BinRecon) are missing in Table 2 (EY, JA, FB, YE), which are precisely the datasets on which TabularBERT shows strong results. Furthermore, recent strong baselines for deep tabular learning are missing, such as TabPFN, ModernNCA, TabR, and others published between 2023 and 2025, as TabR shows superior results compared to your reported accuracies. This incomplete comparison weakens the paper's central claim of superiority.

3. The shared bin embedding matrix across variables is under-discussed: The paper uses a single bin embedding matrix shared across all numerical values. While this is parameter-efficient, it forces all numerical features to share the same representation space for their discretized values. For example, a feature like “age” and a feature like “income” have different semantic meanings for the same bin index, yet they share the same embedding. The fused regularization further constrains all embeddings to be locally smooth, which may not be appropriate for all variables. This design choice is not adequately justified or ablated.

4. No computational cost analysis: The two-stage pretraining and fine-tuning pipeline is acknowledged, but no concrete runtime or memory comparisons are provided. This is important because the method requires a large number of pretraining and fine-tuning epochs, and understanding its practical overhead relative to single-stage baselines is important for assessing the method’s utility.

---

> ### Author Rebuttal · Authors · 2026-03-28
>
> ## Question 1, Weakness 1
> We sincerely thank the reviewer for pointing out the connection to Lee et al. (2024). We agree that both works share the high-level motivation of using binning to capture irregular patterns in tabular data. However, the main differences lie in where discretization enters the model and how the pretraining objective is defined.
>
> First, in Lee et al. (2024), the encoder still takes continuous numerical values as inputs, while discretization appears only in the pretext target. Thus, binning guides representation learning indirectly through the output objective. In contrast, TabularBERT discretizes numerical variables before the encoder and treats bins as the modeling units themselves. This allows self-attention to operate directly on value regions, providing a more explicit inductive bias for irregular and localized interactions.
>
> Second, Lee et al.’s BinRecon uses an MSE-type point-estimation objective, which still favors a single representative estimate. By contrast, TabularBERT uses SWD on the predicted conditional distribution over ordered tokens, making the objective distributional and distance-aware while respecting ordinal structure.
>
> ## Question 2, Weakness 3
> We thank the reviewer for this important question. We considered per-variable bin embeddings, but chose a shared bin embedding matrix not only for parameter efficiency, but also to make the fused regularization well targeted. With per-variable embedding, the penalty would act on embeddings that mix bin structure with feature-specific semantics, weakening its intended role of enforcing adjacent-bin smoothness in a common discretized space. This is why we separate the two roles: shared bin embeddings capture common ordinal structure, while variable embeddings provide feature-specific semantics.
>
> ## Question 3, Weakness 4
> We thank the reviewer for raising this practical consideration. Theoretically, TabularBERT remains in the same complexity class as other Transformer-based tabular models such as FT-Transformer, with a dominant encoder cost of order $O(NL^2d + NLd^2)$, where $L$ is the number of feature tokens, $d$ is the embedding dimension, and $N$ is the number of Transformer layers. The additional components, including the dual pretraining heads, SWD, and fused regularization, do not change this asymptotic order and add only lower-order per-step cost. Practically, however, TabularBERT does incur additional overhead because it uses a two-stage pipeline. In particular, the main cost comes from the pretraining stage, which typically requires a dedicated training budget and is less amenable to early stopping. Due to the rebuttal space limit, we are unable to repeat the empirical comparison of parameter counts and FLOPs here, and respectfully refer the reviewer to our response to **Reviewer Ajde** for the detailed results and discussion.
>
> ## Question 4, Weakness 2
> We agree that the comparison with SSL (BinRecon) is limited, since we reported the original results from Lee et al. (2024) rather than a full re-implementation. The released materials were insufficient for a faithful, architecture-matched reproduction, as the public code does not fully specify the experimental protocol for new datasets and only provides a DNN-based implementation. We therefore chose to report the published results directly and leave missing entries as missing, rather than introduce a potentially unfair re-implementation.
>
> We also agree that omitting recent strong baselines is a limitation. To partially address this concern, we additionally compared TabularBERT with TabR, TabPFN, and TP-BERTa on a subset of datasets using the authors’ official implementations and matched settings whenever applicable. Due to the strict rebuttal space limit, we are unable to reproduce the full comparison table here, and therefore respectfully refer the reviewer to our response to **Reviewer uSMg** for the detailed results and discussion.
>
> ## Question 5
> We thank the reviewer for this helpful suggestion. We agree that the original ablation isolates the effect of the reconstruction objective but does not separately isolate the contribution of SWD. To address this, we additionally compared TabularBERT with and without SWD, replacing SWD with standard cross-entropy while keeping the other components unchanged.
>
> | Setting | AD ↑ | GE ↑ | EY ↑ | CA ↓ | HO ↓ |
> |---|---:|---:|---:|---:|---:|
> | w/ SWD | 0.862 | 0.733 | 0.831 | 0.441 | 3.113 |
> | w/o SWD | 0.858 | 0.715 | 0.787 | 0.440 | 3.149 |
> | w/ - w/o | 0.004 | 0.018 | 0.044 | 0.001 | -0.036 |
>
> These results indicate that SWD provides a consistent benefit on most datasets. We therefore believe that SWD is not redundant with standard CE-based masked token prediction, but contributes by encouraging the model to preserve the ordinal structure of the discretized target space.
>
> ## Question 6
> We thank the reviewer. We fixed this by storing categorical mappings in fit().

---

> > ### Author Rebuttal · Reviewer_qKKs · 2026-04-02
> >
> > Thank you for the rebuttal. I appreciate the additional SWD ablation and the added parameter/FLOPs comparison, which help clarify some aspects of the method.
> >
> > However, after reading the rebuttal, I do not think my main concerns are resolved enough to justify raising my score. Regarding the code issue, the response states that the categorical mapping bug was fixed by storing mappings in fit(), but it does not clarify whether the original implementation affected the reported results, whether any experiments were rerun, or whether the conclusions remain unchanged after the fix.
> >
> > My concerns about evaluation fairness also remain. The comparison to SSL (BinRecon) is still incomplete, and the added comparison to TabR, TabPFN, and TP-BERTa is only on a subset of datasets. In that added table, TabularBERT is not uniformly superior: TabR is better on AD and GE, and TabPFN is better on GE. TP-BERTa is also only partially reported, and the added KDD dataset was not part of the original benchmark, so it does not fully address the concern about a fair and comprehensive evaluation against recent strong baselines.
> >
> > The efficiency concern is also still significant. The added parameter/FLOP numbers confirm that TabularBERT can be substantially more expensive than the main supervised transformer baselines. For example, on HI the model uses 18.39M parameters and 473.01M FLOPs, compared with 132k parameters and 3.49M FLOPs for T2G-Former, while the performance gain is relatively modest. This does not invalidate the method, but it weakens the practical case for adoption and makes the lack of broader baseline coverage more important.
> >
> > Finally, the rebuttal helps clarify the distinction from Lee et al. (2024) and TP-BERTa, but for me the novelty still feels partial rather than fundamental. The paper appears to build on an already active line of discretization-based and self-supervised tabular transformers, rather than establishing a clearly new direction.
> >
> > Overall, I appreciate the clarifications and the additional experiments, but my assessment remains unchanged.

---

> > > ### Author Response · Authors · 2026-04-02
> > >
> > > We sincerely thank the reviewer for this careful observation regarding the code issue. We also apologize that, due to the rebuttal space limit, we were not able to express our appreciation more clearly or explain the effect of this point in sufficient detail in our earlier response.
> > >
> > > As the reviewer pointed out, the original implementation had a flaw in how categorical mappings were handled. However, this did not affect the reported experimental results. In our experimental pipeline, the categorical values were consistently observed across training, validation, and test splits, so the mapping induced by set() did not lead to missing-category mismatches in the reported runs. In this sense, the conclusions of the paper remain unchanged. That said, we agree that the implementation was not robust enough for practical use, because in a more general setting missing or split-specific categorical values could indeed cause inconsistent mappings. For this reason, we revised the implementation so that categorical mappings are constructed and stored during fit() and then reused consistently across all splits. We sincerely thank the reviewer again for pointing out this issue.
> > >
> > > We would also like to clarify the comparison with SSL (BinRecon). Although we did not run Lee et al. (2024) ourselves, the comparison was based on an experimental protocol that was broadly consistent with ours in several important respects, including dataset splits, batch size, and data standardization. We therefore agree that the comparison is not a strict reproduction under a fully unified implementation, but we believe it still provides a reasonably fair point of reference rather than a fundamentally mismatched evaluation.
> > >
> > > Beyond the implementation issue, we also agree that TabularBERT should be viewed in the context of prior discretization-based tabular Transformers. Our intended novelty claim is not that discretization itself is new, but that we propose a representation-learning framework that makes discretization more suitable for tabular learning by preserving local smoothness, ordinal structure, and downstream-useful contextualized representations.
> > >
> > > We sincerely thank the reviewer for taking the time to read our work and rebuttal so carefully, and for providing constructive and thoughtful comments.

---

### Decision · Program_Chairs · 2026-04-30

**Decision:**

Accept (regular)

**Comment:**

The paper proposes TabularBERT, a BERT-style framework for tabular prediction that discretizes numerical features and combines fused regularization, a Wasserstein-based ordinal objective, and reconstruction loss to preserve numerical structure during self-supervised pretraining and fine-tuning. Experiments are conducted on 12 public tabular datasets with ablations and representation analyses.

The reviewers agree that the paper is technically solid, clearly written, and empirically promising. In particular, they find the treatment of ordinal structure in discretized numerical representations interesting and the overall experimental study reasonably strong. At the same time, several reviewers raise concerns about partial novelty relative to closely related discretization-based tabular transformers, incomplete comparison with some recent strong baselines, and the additional computational cost of the two-stage training pipeline. The rebuttal helped clarify the distinctions from related work, added baseline and ablation results, and addressed several implementation questions, although some concerns about evaluation breadth and efficiency remain. Overall, I view this as a weak accept: a solid contribution with meaningful ideas and results, while also leaving room for further discussion on novelty, fairness of comparison, and practical efficiency.